# Elongation roadblocks mediated by dCas9 across human genes modulate transcription and nascent RNA processing

Inna Zukher [1,2,3] ✉, Gwendal Dujardin [1,2], Rui Sousa-Luís[1] & Nick J. Proudfoot [1,3] ✉

Non-cleaving Cas9 (dCas9) is widely employed to manipulate specific gene loci, often with scant regard for unintended transcriptional effects. We demonstrate here that dCas9 mediates precise RNA polymerase II transcriptional pausing followed by transcription termination and potential alternative polyadenylation. By contrast, alternative splicing is unaffected, likely requiring more sustained alteration to elongation speed. The effect on transcription is orientation specific, with pausing only being induced when dCas9-associated guide RNA anneals to the non-template strand. Targeting the template strand induces minimal effects on transcription elongation and thus provides a neutral approach to recruit dCas9-linked effector domains to specific gene regions. In essence, we evaluate molecular effects of targeting dCas9 to mammalian transcription units. In so doing, we also provide new information on elongation by RNA polymerase II and coupled pre-mRNA processing.

Mammalian protein-coding genes are selectively transcribed by RNA polymerase II (Pol II), with their transcription units often extending for hundreds of kilobases. The resulting nascent transcript is cotranscriptionally spliced and polyadenylated. This suggests that regulating Pol II elongation rates should impact both RNA processing and transcript levels. Indeed, Pol II pausing and changes in elongation rate are known to alter alternative splicing (AS)[1], 3′ end RNA processing[2] and transcription termination[3]. Moreover, both exons and transcription end regions are associated with Pol II accumulation, which is thought to facilitate pre-mRNA processing[4,5].

Here, we describe the effects of sequence-specific gene targeting to manipulate Pol II progression. In recent years, CRISPR–Cas9 gene clusters, which are a natural part of immunity systems in many bacteria, have been repurposed as powerful genetic engineering tools[6]. Thus, modified CRISPR–Cas9 system, derived from *Streptococcus pyogenes*, requires a single-guide RNA (sgRNA) to recruit Cas9 nuclease to 23-nucleotide DNA genomic targets (Fig. 1a). Nuclease generates a double-stranded break, affording effective genome editing[7–9]. Notably, the dCas9 mutant (Cas9[D10A/H840A]) lacks endonuclease activity but retains sgRNA-guided DNA-binding activity[7] and can be fused to

effector domains to increase experimental versatility[10]. In particular, domains activating transcription (CRISPRa systems) or interfering with it (CRISPRi systems) can be employed[11]. Although a wide range of sgRNA libraries are available to use these systems in multiple organisms, little is known about the transcriptional consequences of dCas9 targeting. When bound directly to the transcription start site (TSS), dCas9 suppresses gene expression by physically blocking access of transcription factors to promoters[12,13]. Instead, when bound downstream of the TSS, dCas9 blocks progression of the Pol II elongation complex (EC)[12]. Although over the TSS region its suppression effect is orientation independent, at downstream positions dCas9 only inhibits gene expression when targeted to the non-template (NT) strand[12,13].

Here, we compare the molecular consequences of dCas9 targeting across human protein-coding genes. When dCas9 binds actively transcribed DNA, it creates a transcriptional roadblock that induces pausing of the Pol II EC, followed by transcription termination. In regions lacking active polyadenylation signals (PAS), this results in premature termination, followed by degradation of non-polyadenylated nascent transcript, and thus represses gene expression. Instead, targeting dCas9 downstream of the PAS may not alter gene expression but

[1]Sir William Dunn School of Pathology, University of Oxford, Oxford, UK. [2]These authors contributed equally: Inna Zukher, Gwendal Dujardin. [3]These authors jointly supervised this work: Inna Zukher, Nick J. Proudfoot. ✉e-mail: inna.zukher@path.ox.ac.uk; nicholas.proudfoot@path.ox.ac.uk

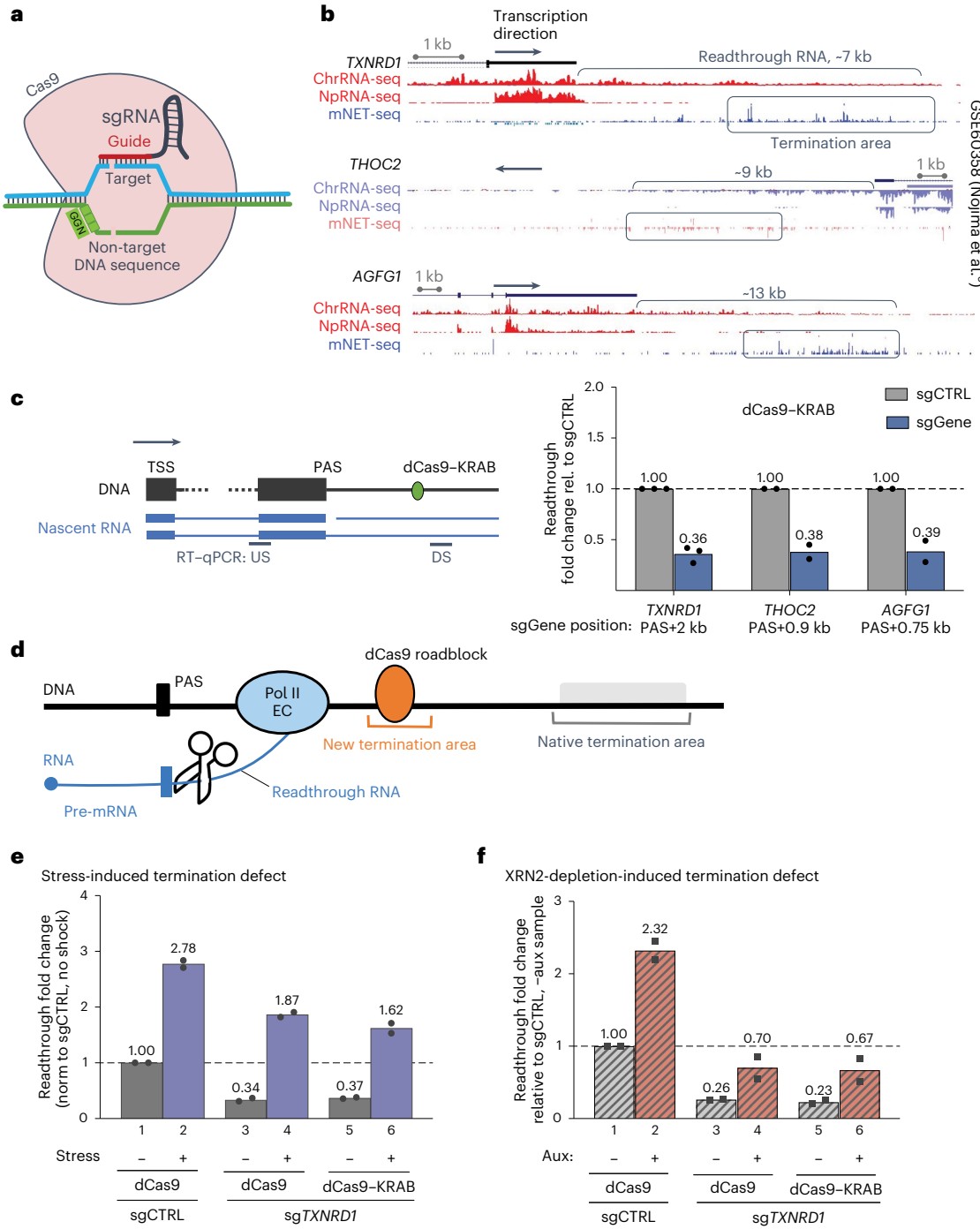

**Fig. 1 | CRISPRi targeted downstream of a gene prevents transcriptional readthrough. a**, Diagram shows *S. pyogenes* Cas9 (SpCas9) binding to the 23-nucleotide target site, with the sgRNA guide region (red) base pairing to the 20-nucleotide DNA target sequence (blue strand) and the three-nucleotide NGG PAM on the opposite strand (in green). Double-stranded DNA cleavage is also shown. Note that all details remain the same for dCas9 mutant binding, but DNA is not cleaved. **b**, Screenshots from the University of California Santa Cruz (UCSC) browser show representative chromatin (Chr) and nucleoplasm (Np) RNA-seq and total Pol II mNET-seq profiles (GSE60358 dataset[5]). **c**, HeLa cells were transfected to express dCas9–KRAB and non-targeting (sgCTRL, gray bars) or 3′-end gene-specific (sgGene, blue bars) sgRNA, targeting at 1.2 kb, 0.9 kb or 0.75 kb downstream from the PAS of *TXNRD1*, *THOC2* or *AGFG1*, respectively. Transcriptional readthrough for the targeted gene was analyzed by random-primed RT followed by qPCR (see diagram on left) and expressed as a ratio between RT–qPCR signal upstream (US) and downstream (DS) of the CRISPRi block (downstream/upstream ratio). Data were normalized to the readthrough

level in control cells. Ratio values <1 denote less readthrough than in the sgCTRL sample. Data from biologically independent experiments are presented with the mean value indicated on top of the bar. Rel., relative. **d**, Diagram depicting the position of the dCas9 roadblock downstream of a gene with the expected shift from a native (gray) to an induced (orange) termination area. **e**, HeLa cells were transfected to express dCas9 or dCas9–KRAB and either sgCTRL or *TXNRD1*-specific sgRNA species and subjected to osmotic shock for 1 h (blue bars) or no treatment (gray bars). Transcriptional readthrough was analyzed as described above. Data were normalized (norm) to the control, no stress sample; values >1 denote more readthrough. Data from *n* = 2 biologically independent replicates are shown with the mean value indicated on top of the bar. **f**, HCT116 TIR1 XRN2-AID cells were transfected with the same constructs as in **e** and treated with auxin (aux) for 2 h to induce XRN2 depletion (striped red bars) or control (striped gray bars). Transcriptional readthrough was analyzed as described above. Data were normalized to the control, no auxin sample. Data from *n* = 2 biologically independent replicates are shown with the mean value indicated on top of the bar.

suppresses readthrough transcription. We also show that dCas9 only promotes Pol II pausing when targeted to the NT strand, effectively operating as a molecular valve to enforce unidirectional transcription. Overall, our results define new parameters to determine the optimal positions for placement of dCas9 that will elicit either maximal gene repression or minimal transcription disturbance.

## Results

### CRISPRi suppresses transcriptional elongation

As described in the Methods, we employed a combination of targeting guides and dCas9 fusion with repressive Krüppel-associated box domain (dCas9–KRAB) to promote localized chromatin suppression[13] of the *DPH2* TSS region. Notably, our CRISPRi system induced a 40-fold reduction in steady-state *DPH2* mRNA levels (Extended Data Fig. 1a). We then tested the effects of dCas9–KRAB targeting in the 3′ end regions of three selected protein-coding genes[5,14] that display a wide transcription-termination zone, multiple kilobases beyond the terminal PAS. Screenshots for the 3′ regions of these genes show previously published RNA-seq analyses of chromatin and nucleoplasmic RNA, corresponding to nascent (chromatin) and processed (nucleoplasmic) transcripts, respectively. Additionally, mammalian native elongating transcript sequencing (mNET-seq) analysis shows Pol II-associated nascent RNA 3′ ends (Fig. 1b).

For each gene, we used a set of four to six NT sgRNA species, together targeting 200–400 bp within 1–2 kb downstream of the PAS. Reverse transcription followed by quantitative polymerase chain reaction (RT–qPCR) analysis revealed almost threefold suppression of transcriptional readthrough with the specific sgRNA species (Fig. 1c). These data demonstrate the potency of CRISPRi in both blocking promoter activity when targeted to gene 5′ ends but also suppressing transcriptional elongation when targeted downstream of the gene body. A plausible mechanism for this effect is premature termination (Fig. 1d). We therefore tested whether external factors, such as osmotic stress[15], known to induce a termination defect also antagonize CRISPRi effects. Notably, osmotic stress induced almost a threefold increase in transcriptional readthrough downstream of *TXNRD1* in either control- or CRISPRi-treated cells (Fig. 1e). Both dCas9 alone and dCas9–KRAB gave similar effects, indicating that CRISPRi-dependent readthrough suppression is still sensitive to osmotic stress. This suggests that the same machinery is required and that the CRISPRi effect depends solely on dCas9 DNA binding. We also tested the role of 5′–3′ exoribonuclease 2 (XRN2) in the CRISPRi-induced termination process. Pol II transcription termination normally occurs via a 'torpedo' mechanism, with XRN2 binding to the phosphorylated nascent transcript 5′ end generated upon PAS cleavage (Fig. 1d). XRN2 then processively digests the transcript, reaches the EC and displaces it from the DNA template[3]. To inhibit this mechanism, we used an XRN2-AID cell line that allows rapid auxin-induced XRN2 degradation[16]. Due to basal degradation occurring even in the absence of auxin induction, XRN2-AID levels are reduced in this cell line[17], leading to some loss of termination efficiency (Extended Data Fig. 1b). Nevertheless, XRN2 depletion increased *TXNRD1* readthrough transcript levels twofold (Fig. 1f). Cells transfected to express sg*TXNRD1* were more resistant to readthrough stimulation than control cells under the same conditions. As with osmotic stress, we observed no difference between the effects of dCas9 and dCas9–KRAB constructs.

Overall, these results demonstrate that CRISPRi-mediated termination can still be antagonized by natural antitermination effects.

### The roadblock effect is unchanged by repressive chromatin marks

We have previously observed that termination is often associated with Pol II pausing[5,18,19]. Furthermore, pausing may result in epigenetic changes to chromatin structure such as the acquisition of histone 3 lysine 9 (H3K9) dimethylation (H3K9me2) or trimethylation (H3K9me3)

and associated heterochromatin protein 1 γ (HP1γ) recruitment[19]. Possibly, CRISPRi-mediated transcription roadblocks induce such repressive epigenetic chromatin structures. We therefore employed both the sg*TXNRD1* set targeting 2 kb downstream of the PAS (DS3) as used above and a second set targeting 1.3 kb downstream of the PAS (DS2). Chromatin immunoprecipitation followed by qPCR (ChIP–qPCR) was carried out across the region downstream of the *TXNRD1* PAS using antibodies for HP1γ or Pol II with Thr4 phosphorylation in C-terminal domain (T4P CTD) of the large subunit. This T4P mark is associated with transcription termination[14,20–22]. We consistently observed HP1γ signal peaks for both dCas9 and dCas9–KRAB just upstream of their respective sgRNA target sites (Fig. 2). Pol II T4P signal peaks were also observed at the matching positions, indicating Pol II stalling and transcription termination at target sites.

HP1γ can be recruited to H3K9me2 or H3K9me3 marks[23] but also can be associated with active Pol II ECs[24,25]. The KRAB domain recruits histone methyltransferases and thus locally increases repressive H3K9me3 marks in the vicinity of CRISPRi target sites[26]. To determine whether HP1γ was brought to Pol II T4P stalling sites by H3K9me3, we compared ChIP–qPCR profiles for cells bearing dCas9 and dCas9–KRAB constructs (Fig. 2). Indeed, with chromatin from cells expressing dCas9–KRAB, we detected notable H3K9me3 signal. Interestingly, it was specifically depleted directly at the target sites (DS2 and DS3), suggesting steric restriction for histone methylation at the sites occupied by dCas9–KRAB. Even though H3K9me3 spreads over a wide region upstream and downstream of the dCas9–KRAB block, Pol II and HP1γ patterns remained similar to those of chromatin from dCas9-expressing cells. This argues that CRISPRi-dependent polymerase pausing is not caused by the H3K9me3 mark and its associated complexes but rather by dCas9 roadblocks on the DNA template. These results are consistent with the RNA data described above (Fig. 1e,f), where dCas9–KRAB and dCas9 displayed similar readthrough suppression. In effect, our findings differ from previously described promoter-targeting CRISPRi systems, where dCas9–KRAB enhances the suppression effect[26]. Instead, when dCas9–KRAB binding induces widespread H3K9me3 methylation downstream of the PAS, Pol II elongation is unaffected. We predict that, once processive transcription elongation is established, it becomes insensitive to local chromatin structure, unlike the transcriptional initiation and early elongation stages.

Another potential HP1γ-recruiting feature is H3K9me2, which has previously been described as a termination trigger[19]. H3K9 is dimethylated by the G9a–GLP histone methylase complex, which can be specifically inhibited by UNC-0638 (ref. 27) or BIX-01294 (ref. 28) compounds. Western blot analysis confirmed that both drugs reduced H3K9me2 levels in HeLa cells (Extended Data Fig. 2a). However, this decrease in H3K9me2 did not affect the intensity of Pol II T4P, HP1γ or dCas9 ChIP signals at the CRISPRi target sites (Extended Data Fig. 2b).

In sum, our data demonstrate that CRISPRi targeting downstream of the PAS induces Pol II pausing upstream of the DNA-bound roadblock, followed by specific Thr4 CTD phosphorylation, that eventually triggers termination. This CRISPRi-induced termination process likely relies on the torpedo mechanism, in view of its sensitivity to XRN2 exonuclease depletion (Fig. 1f). However, this artificial termination process does not require H3K9me2 or H3K9me3 and is not further stimulated by their presence. We also show that additional HP1γ observed in the termination site is not recruited to H3K9me2 or H3K9me3 but is likely associated with the EC.

### Asymmetry of the CRISPRi roadblock

The sgRNA species described above were all designed in antisense orientation to target the NT DNA strand. Indeed, CRISPRi with sgRNA species in sense orientation, targeting the template (T) strand, have almost no gene-repression effect outside of the TSS region[12,29], suggesting that the elongation roadblock effect is asymmetric. There are two possible Pol II–dCas9 collision scenarios. Either elongating Pol II

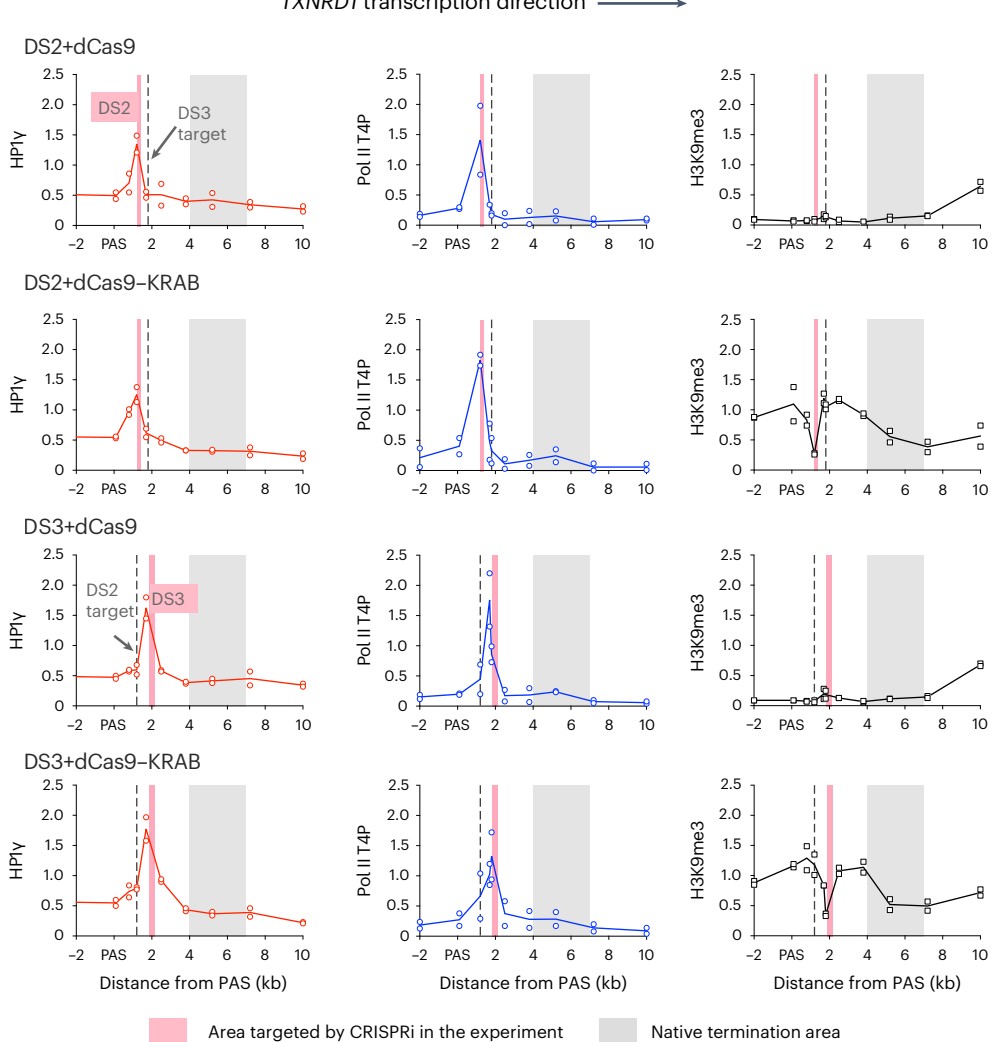

**Fig. 2 | CRISPRi and CRISPRi-KRAB induce Pol II and HP1γ accumulation immediately upstream of sgRNA target sites independent of repressive chromatin marks.** HeLa cells were transfected with constructs encoding dCas9 or dCas9–KRAB and *TXNRD1* DS2 or DS3 sgRNA sets, subjected to ChIP–qPCR with antibodies specific either to HP1γ (red graphs), the Pol II T4P CTD form ('Pol II T4P', blue graphs) or H3K9me3 (black graphs). DS2 and DS3 target sites are centered at 1.3 and 2.0 kb downstream of the PAS, respectively. HP1γ and Pol II T4P signal in each sample was normalized to immunoprecipitate (IP)/ input enrichment for control *MYC* 3′ end DNA qPCR signal in this sample, and H3K9me3 signal was normalized to centromeric DNA qPCR signal. Data from $n = 2$ biologically independent replicates are presented.

first encounters the NGG (PAM, protospacer-adjacent motif) end of the target site, when the antisense guide base pairs with the NT strand, or the guide RNA end, when the sense guide base pairs with the T strand (Fig. 3a). To further explore this roadblock asymmetry, we designed constructs that target the T strands of the *TXNRD1* DS2 and DS3 regions as above (Fig. 2): DS2-T and DS3-T. In this experiment, we used the antibody against N-terminal domain of RNA polymerase II large subunit (RPB1), which immunoprecipitates total Pol II independently of its CTD modifications. Similarly to profiles observed with terminating T4P CTD Pol II (Fig. 2a), Pol II NTD ChIP–qPCR profiles have signal peaks just upstream of the target sites (Fig. 3b). Remarkably, Pol II NTD ChIP–qPCR profiles for T constructs were almost indistinguishable from those of control cells, indicating that T-targeted dCas9 did not present an efficient elongation roadblock even though dCas9 ChIP–qPCR signal in chromatin from cells with T constructs was the same or higher than that for NT constructs (Fig. 3c). This suggests that the dCas9 roadblock operates as a molecular valve, allowing transcription in one direction only. It allows transcription by the EC approaching from the PAM-distal end but not the PAM-proximal end. We suggest that this asymmetry is

the main cause of the reported T-targeting inefficiency for CRISPRi gene repression.

Cryo-electron microscopy analysis of dCas9–sgRNA in complex with over 50 bp-long double-stranded DNA template has been described[30]. There, the DNA template contained a 23-bp target site flanked by 16-bp upstream and downstream sequences (Fig. 3d). The PAM-upstream DNA fragment is not observed in the structure, which implies that it is either freely rotating or disordered. By contrast, the density for the PAM-downstream 16-bp fragment is clearly present as a double-helical protuberance (Fig. 3d). This suggests that dCas9– sgRNA binding to the target sequence creates a rigid DNA structure downstream of the PAM, even though this DNA is not in direct contact with dCas9. The presence of multiple protein–RNA–DNA contacts in the PAM-proximal region agrees well with the view that the PAM and the protospacer 'seed' region of 8–12 nucleotides adjacent to it are crucial for target recognition[7,31]. Considering that Pol II pausing is only observed with NT targeting (Fig. 3b), we argue that dCas9 presents a stronger physical barrier for either EC progression or DNA melting when approached by the EC from the PAM-proximal side. This effect

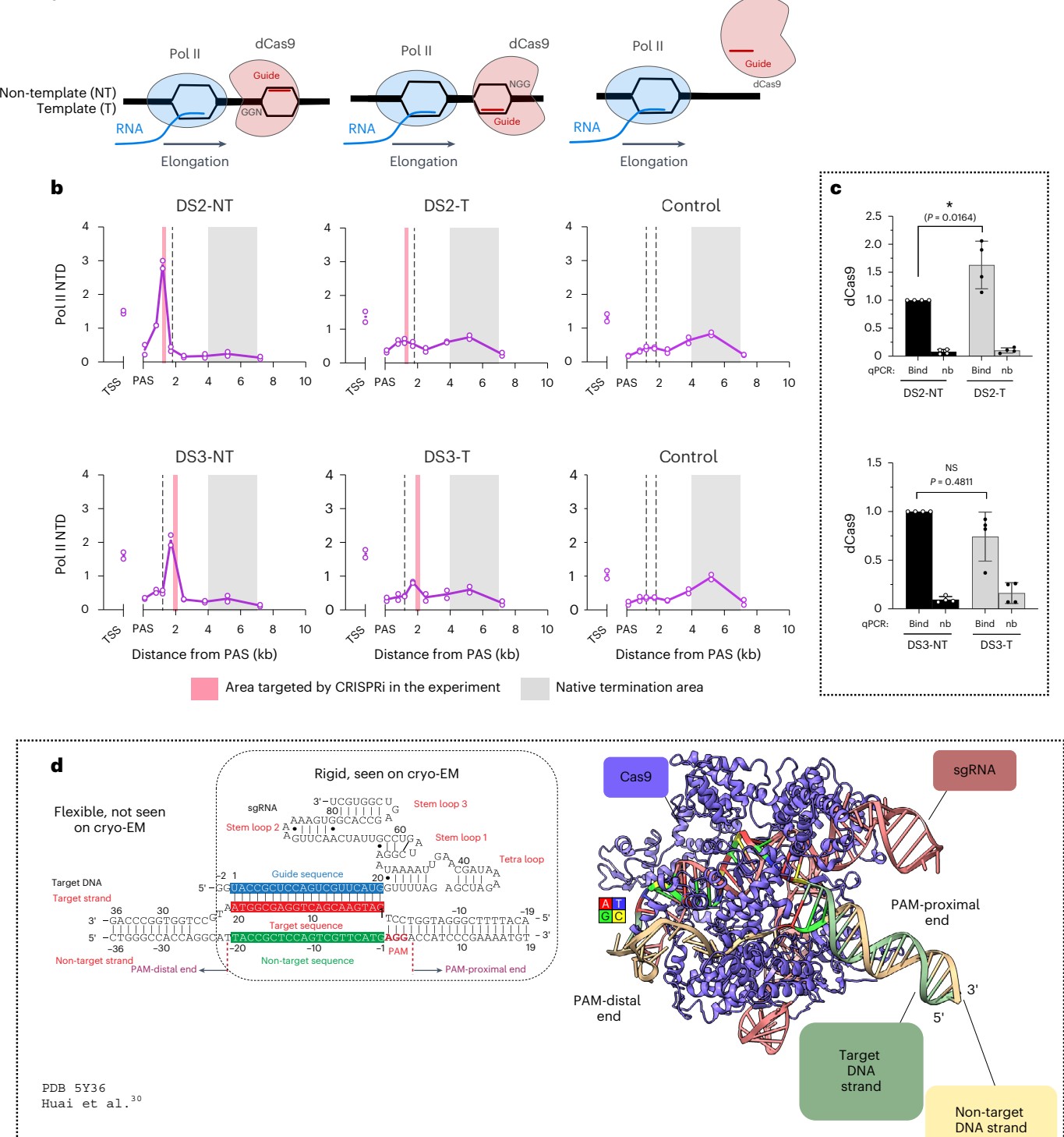

**Fig. 3 | The CRISPRi roadblock effect is strand specific. a**, Diagram depicts T and NT targeting by guide RNA. **b**, Pol II NTD ChIP–qPCR with HeLa cells transfected to express dCas9 and sgRNA species targeting either the NT or T DNA strand in *TXNRD1* DS2 or DS3 regions or control sgRNA. Signal in each sample was normalized to IP/input enrichment for control *MYC* 3′-end DNA qPCR signal. Data from *n* = 2 biologically independent replicates are shown. **c**, Chromatin from HeLa cells transfected with constructs targeting either the NT or T DNA strand of *TXNRD1* DS2 or DS3 regions used for ChIP–qPCR with anti-FLAG antibody, recognizing the FLAG tag on dCas9. The 'binding' qPCR product is specific to the respective block position; 'nb' denotes that a primer pair outside of the binding site controls for nonspecific ChIP signal. Signal was normalized to signal from binding site DNA in the respective NT sample. Data are presented as mean values ± s.d. for *n* = 4 biologically independent replicates; one-way ANOVA with Tukey's multiple-comparison test. NS, not significant. **d**, Adapted from Fig. 1 in ref. 27. Diagram shows target DNA and sgRNA used for the ternary complex; potential Watson–Crick and non-Watson–Crick base pairs in sgRNA and DNA are indicated by lines and dots, respectively. On the right is shown the cryo-electron microscopy (cryo-EM) structure of the SpCas9–sgRNA–DNA ternary complex (PDB 5Y36). dCas9, blue; sgRNA, red; target DNA strand, green; non-target DNA strand, yellow. Protospacer and PAM (AGG) are colored according to the DNA sequence (red, A; green, G; yellow, C; blue, T).

likely correlates with stronger protein binding and the observed DNA rigidity on this side of the binding site. The apparent DNA flexibility and fewer dCas9-sgRNA complex contacts with the DNA on the PAM-distal side may afford easier EC progression through the block upon T targeting. These data strongly suggest that T targeting should be considered for applications in which dCas9 is employed to recruit enzymatic activities or imaging tags, as it creates minimal disturbance to transcription at the target site. The molecular valve feature, or polarity of the dCas9 roadblock toward EC progression, has been recently demonstrated in a bacterial in vitro system by single-molecule assays with the *Escherichia coli* RNA polymerase EC[32]. Consistently, dCas9 binding restricts translocation only for the EC approaching it from the PAM-proximal side.

### Elongating Pol II progression is modulated by CRISPRi roadblock

We demonstrate above that CRISPRi targeted downstream of genes acts to pause ECs, suppress transcription readthrough and induce transcription termination. To further evaluate how this effect varies depending on target site position within the transcription unit, we compared a set of CRISPRi constructs targeting the NT DNA strand of the *TXNRD1* PAS-proximal regions either upstream (untranslated region (UTR); −1.4 kb from the PAS) or downstream (DS1, DS2 and DS3; +0.5–2 kb) of the PAS as well as within the 22-kb-long intron 2 (in2; −52 kb) (Fig. 4a and Extended Data Fig. 3a). To directly measure the effect of CRISPRi targeting throughout *TXNRD1* on transcription elongation, we generated total (NTD) and terminating (T4P) Pol II ChIP–qPCR profiles for transfected cells (Fig. 4b). Control cells with the CRISPRi CTRL construct yielded ChIP–qPCR profiles with a wide peak 4–7 kb downstream of the PAS for both anti-Pol II NTD and T4P antibodies (Fig. 4b), in agreement with previously observed mNET-seq and chromatin RNA-seq profiles for this gene (Fig. 1b). By contrast, all five *TXNRD1*-specific constructs induced a prominent Pol II peak just upstream of the respective target site, again both for Pol II NTD and T4P profiles. Notably, chromatin from the cells transfected with CRISPRi in2 and UTR constructs exhibited strong Pol II peaks at dCas9-binding sites within the gene, but their 3′ downstream profiles remained unchanged from those of the control samples, with a wide peak 4–7 kb downstream of the PAS. CRISPRi DS1 induced a double-peak ChIP–qPCR profile, with one peak located immediately upstream of the block and the other over the native termination zone. Finally, cells transfected with CRISPRi DS2 or DS3 displayed pronounced Pol II peaks at 1.2 or 1.8 kb, just upstream of their respective blocks, while the 4–7-kb native termination peak was lost. Interestingly, the Pol II peak signal induced by PAS-upstream in2 and UTR blocks was much more pronounced than that for blocks targeted downstream. This increase in Pol II signal is matched by a tenfold stronger dCas9 binding, observed for in2 and UTR cells in dCas9 ChIP–qPCR analysis (Extended Data Fig. 3b). This suggests that Pol II pausing intensity is proportional to dCas9-binding efficiency. Such a difference in dCas9 binding throughout the gene may relate to more active transcription within the gene body, which could render the DNA targets more accessible.

Analysis of nascent and processed RNA levels in transfected cells demonstrates that roadblocks targeting in2 and UTR reduce *TXNRD1* mRNA level (Fig. 4c) but do not affect readthrough transcription ratio

(Fig. 4d). By contrast, upon targeting DS2 and DS3 regions, gene expression remains unchanged (Fig. 4c), suggesting that a shifted termination profile downstream of the PAS does not interfere with Pol II turnover. CRISPRi DS1, targeting the region downstream of but very close to the PAS, reduces both mRNA levels and readthrough transcription. Surprisingly, the intensity of dCas9 and Pol II signal at the target site did not predict the magnitude of RNA effect. Thus, CRISPRi in2, UTR and DS1 decrease *TXNRD1* expression to 30–40% of the control cell level (Fig. 4c), even though dCas9 signals at in2 and UTR sites are 10–20-fold stronger (Extended Data Fig. 3b) and induce stronger Pol II stalling (Fig. 4a) than at DS1. However, gene-repression levels in CRISPRi in2-, UTR- and DS1-transfected cells correlate with decreased Pol II ChIP–qPCR signal downstream of the PAS (Extended Data Fig. 3c). This suggests that, upon intragenic targeting, a similar fraction of Pol II complexes that initiate transcription ultimately reach the end of the gene. The slightly increased Pol II signals downstream of the PAS for CRISPRi DS2 and DS3 may represent delayed Pol II release upon CRISPRi-induced transcription termination.

In summary, these data demonstrate that artificial termination outcome is defined by the local context of dCas9 binding. CRISPRi targeted downstream of the PAS induces EC stalling, followed by Thr4 CTD phosphorylation and Pol II release from the DNA template upstream of the native termination zone. This process, however, does not alter gene expression. By contrast, CRISPR interfering with the EC upstream of the PAS results in premature artificial termination and reduced gene expression.

### An internal roadblock derepresses a nested transcription unit

To characterize genome-wide effects of CRISPRi roadblocks, we performed RNA-seq analysis of chromatin-bound RNA (Chr) and polyadenylated (polyA+) RNA from cells with control or *TXNRD1* sgRNA species targeting either within (in2) or downstream (DS2) of the gene body (Fig. 5). As with the observed increase in Pol II RT–qPCR signal upstream of the CRISPRi target sites described above (Fig. 4b), we detected a local increase in chromatin-bound RNA signal in the same areas (Fig. 5a), especially prominent for CRISPRi DS2. There is also a clear reduction in the signal downstream of the target site, reflecting decreased readthrough transcription for CRISPRi DS2 cells, confirming the above RT–qPCR data (Fig. 4d). Quantitation of the chromatin RNA-seq read count ratio between CTRL and in2 or DS2 samples further demonstrated decreased RNA signal downstream of the target site (Fig. 5a), as also shown in replicates (Extended Data Fig. 4a). Notably, from a selection of genes of similar length and expression, none demonstrated this behavior (Extended Data Fig. 4b), suggesting that the result is specific to transcription suppression by the CRISPRi roadblock.

Differential expression (DE) analysis of polyA+ RNA-seq data confirmed that *TXNRD1* expression is unchanged upon DS2 targeting and reduced upon in2 targeting. Interestingly, in addition to *TXNRD1* suppression in CRISPRi in2 cells, DE analysis also detected activation of *EID3* (Fig. 5b and Extended Data Fig. 4c), also seen in the polyA+ RNA-seq profiles (Fig. 5c). Analysis of the reads crossing exon–exon junctions (Fig. 5c and Extended Data Fig. 4d) did not detect any *EID3* reads overlapping with the *TXNRD1* exons, suggesting that *EID3* is an independent intronless gene nested within the large in2 of *TXNRD1*.

---

**Fig. 4 | CRISPRi effects throughout *TXNRD1*. a**, Scheme depicting dCas9 targets throughout *TXNRD1*. CTRL, control; nt, nucleotide. **b**, HeLa cells were transfected with dCas9 constructs for non-targeting (control) or *TXNRD1*-specific sgRNA species as shown in Fig. 4a. ChIP–qPCR analysis was performed with antibodies specific to Pol II T4P (blue graphs) or Pol II NTD (purple graphs). Signals were normalized to maximum enrichment over input downstream of the *TNXRD1* PAS. The number of biologically independent replicates is shown on each graph. For *n* = 2, individual replicates are presented; for *n* > 2, mean values ± s.d. Arrows show the direction of transcription. Term., termination. **c**, RNA was extracted from the cells transfected as in Fig. 4b and analyzed by oligo-dT RT–qPCR, and

*TXNRD1* expression was normalized to that of *DPH2*. Data from biologically independent replicates are shown with the mean value indicated on top of the bar. For experiments with *n* ≥ 3, one-way ANOVA with Tukey's multiple-comparison test shows differences from the control sample. **d**, Transcriptional readthrough was analyzed by RT–qPCR as in Fig. 1c and normalized to that of control cells. Data from biologically independent replicates are shown with the mean value indicated on top of the bar. For experiments with *n* ≥ 3, one-way ANOVA with Tukey's multiple-comparison test shows differences from the control sample.

*EID3* is specifically expressed in the testis, and its position within *TXNRD1* is conserved between human and mouse genomes[33], underlying its likely functionality. ECs initiated at the *TXNRD1* TSS presumably limit transcription initiation at the downstream *EID3* TSS through transcriptional interference[34,35]. However, when upstream transcription is suppressed by CRISPRi in2 roadblock, interference is reduced and *EID3* TSS becomes active. It is notable that similar internally initiated transcription units may be overlooked in CRISPRi experiments, as they

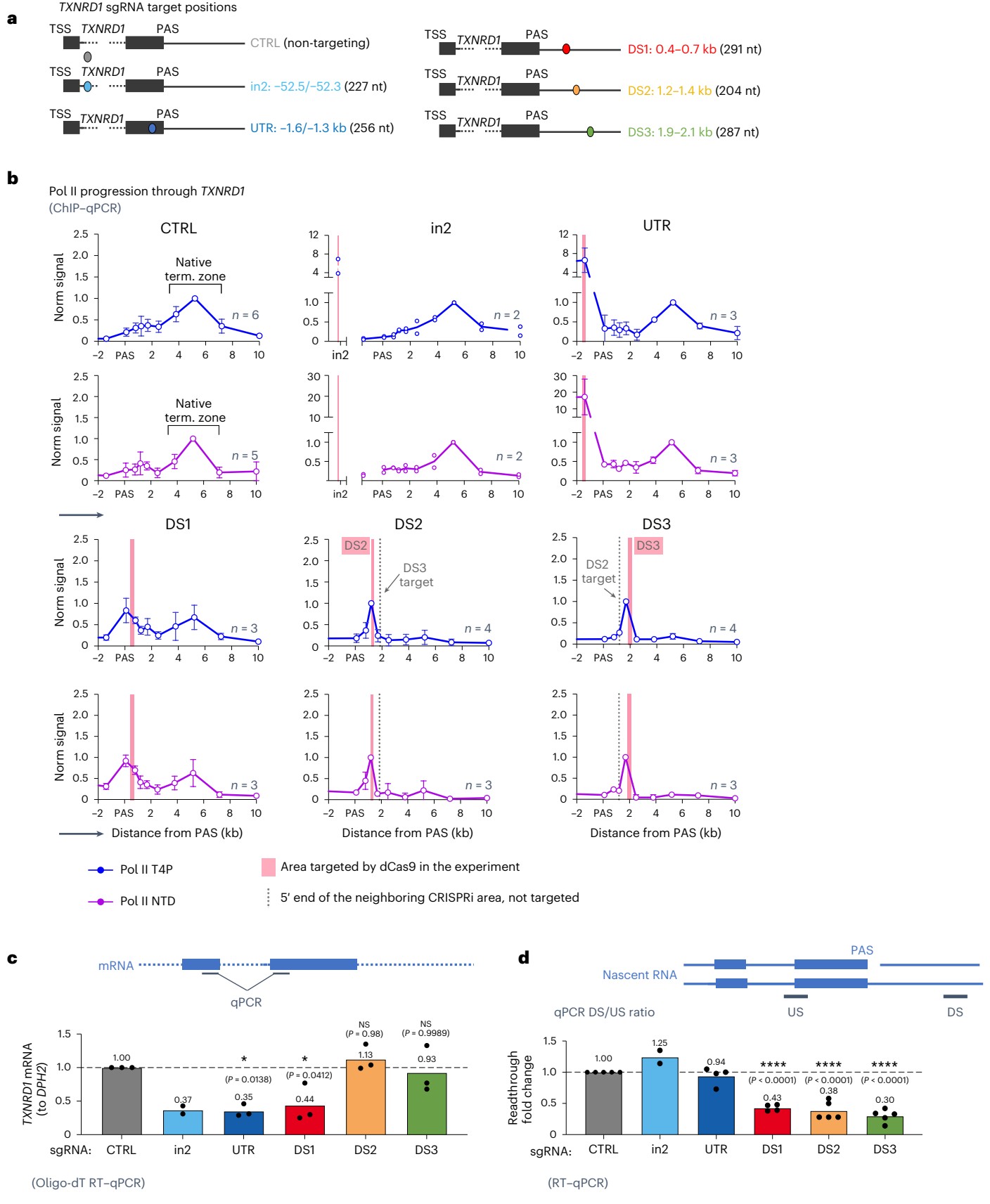

**b** Pol II progression through *TXNRD1* (ChIP–qPCR)

Pol II T4P
Pol II NTD

Area targeted by dCas9 in the experiment

5' end of the neighboring CRISPRi area, not targeted

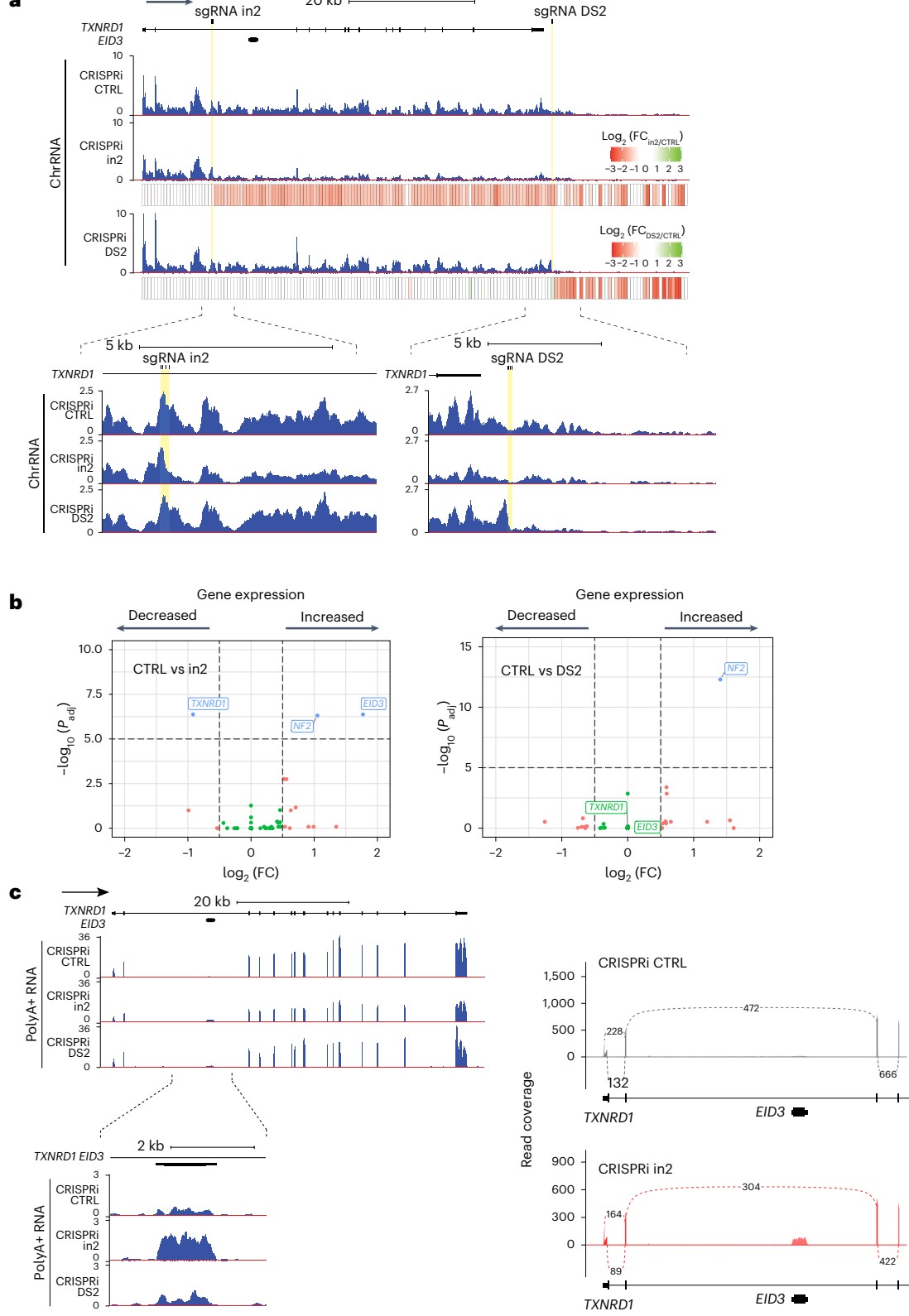

**Fig. 5 | Genome-wide effects of gene-specific CRISPRi targeting.**
**a**, Screenshots from the UCSC browser show representative chromatin RNA-seq profiles for *TXNRD1*. The y scale represents read counts per million mapped reads. Heatmaps below show binning analysis of profiles: each 500-bp bin on the heatmap represents the ratio of read counts in the bin for the in2 or DS2 sample divided by read counts in the control sample; color of the bin shows log₂ (fold change) (log₂ (FC)) for statistically significant differences. **b**, Volcano plot of

polyA+ RNA-seq DE analysis (*n* = 3 biologically independent replicates). *P*~adj~, adjusted *P* value. **c**, Screenshots from the UCSC browser show representative polyA+ RNA-seq profiles for *TXNRD1* and nested *EID3*. The y scale represents read counts per million mapped reads. Sashimi plots of *TXNRD1* splicing are also shown. Numbers on the connecting lines show the average read count supporting each exon–exon junction for *n* = 3 biologically independent polyA+ libraries.

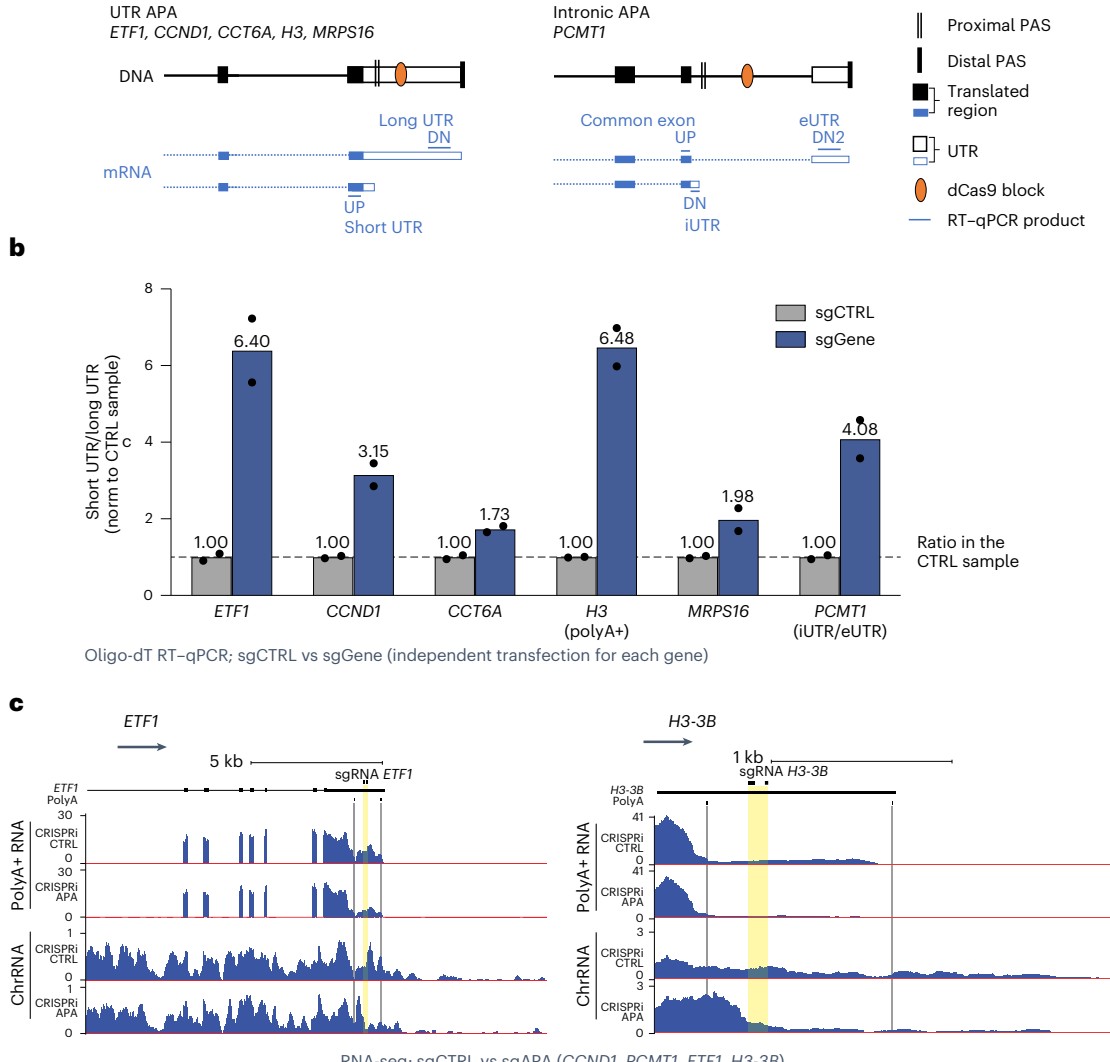

**Fig. 6 | Regulation of APA by modulation of Pol II pausing. a**, Diagram showing CRISPRi targeting and RT–qPCR products used to detect APA isoforms, for *PCMT1* exonic UTR (eUTR) and intronic UTR (iUTR) are shown. **b**, HeLa cells were transfected with dCas9 constructs encoding non-targeting (gray, control) or gene-specific (blue, gene) sgRNA species. Distal and proximal PAS usage is estimated as a ratio of 'up' to 'down' RT–qPCR product for the targeted gene; data were normalized to APA usage in the respective control samples. Data from *n* = 2 biologically independent replicates are shown with the mean value indicated on top of the bar. **c**, Screenshots from the UCSC browser show representative chromatin and polyA+ RNA-seq profiles of HeLa cells transfected with CRISPRi control or APA constructs. *ETF1* and *H3-3B* are shown.

would escape detection by DE analysis, which relies on gene annotation. We suggest that manual inspection of polyA+ RNA-seq profiles in the proximity of CRISPRi targets is important to detect secondary gene-activation effects.

We note that DE analysis revealed no alteration in gene expression outside of the *TXNRD1* locus, emphasizing CRISPRi target specificity (Fig. 5b and Extended Data Fig. 4c). However, for one gene, *NF2*, a statistically significant 2–3-fold increase in expression was observed in CRISPRi in2 and DS2 versus control libraries. We suggest that *NF2* is nonspecifically activated by Cas9–sgRNA binding to unrelated genomic targets.

## Manipulating alternative polyadenylation by CRISPRi
We demonstrate above, by RT–qPCR and chromatin RNA-seq analyses, that CRISPRi promotes targeted EC pausing. This pausing could, in turn, slow down EC progression upstream of the pausing site, giving additional time for the recognition of RNA-processing signals, such as alternative polyadenylation (APA) or AS sites. Indeed, a decreased elongation rate is known to alter alternative APA profiles in *Drosophila* cells under certain conditions[36]. Furthermore, a stably expressed dCas9 was recently used to manipulate APA in endogenous human genes[37].

A set of genes that have been previously reported to display APA (*ETF1*, *CCT6A*, *H3-3B*, *MRPS16* and *PCMT1*)[38] were used as targets for two to three sgRNA species positioned just downstream of the alternative proximal PAS (Fig. 6a). We also tested the previously published APA-modifying *CCND1* sgRNA[37] using our CRISPRi system. The abundance of these mRNA species with alternative UTRs was then measured by RT–qPCR. Targeting CRISPRi upstream of the distal PAS increased proximal UTR PAS usage 2–6-fold as compared to that of control cells (Fig. 6b). Notably, the results we observed in transiently transfected HeLa cells for *CCND1* matched well with previously published data for the stable HEK293T-dCas9 cell line (Fig. S1B in ref. 37), with short UTR mRNA isoform levels increasing about threefold in both cases.

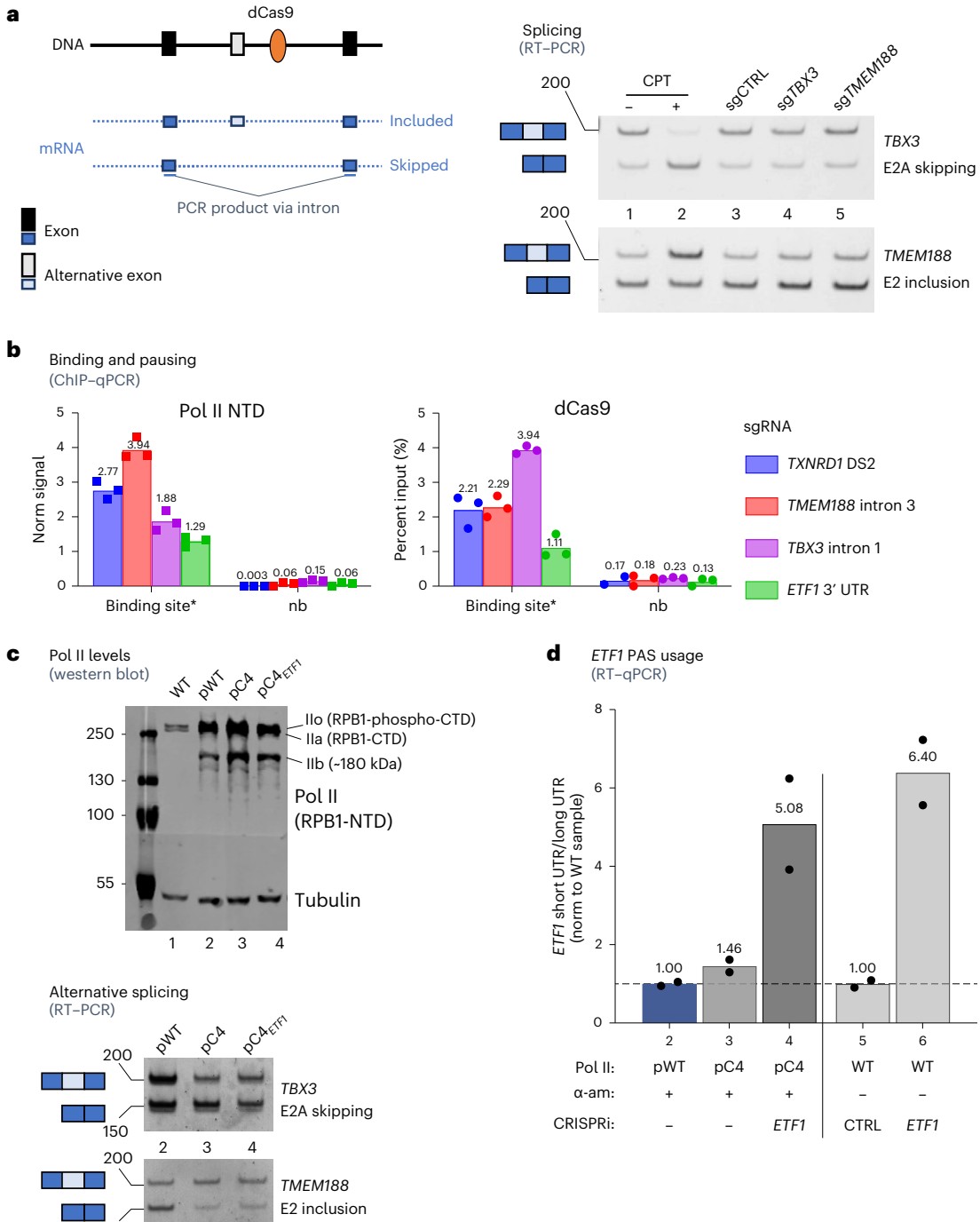

**Fig. 7 | Regulation of APA and AS by slow Pol II and pausing. a**, RT–PCR analysis of RNA from HeLa cells treated with CPT (±CPT) or transfected to express CRISPRi control, *TBX3* or *TMEM188*. Representative endpoint RT–PCR product electropherogram (of *n* = 3 biologically independent replicates). Size marker band positions are shown on the left (length in nucleotides). Top and bottom bands correspond to RT–PCR product from mRNA with included and skipped exons, respectively, as shown in the scheme. **b**, ChIP–qPCR with chromatin from cells transfected to express CRISPRi *TXNRD1* DS2, *TMEM188* intron 3, *TBX3* intron 1 or *ETF1* 3′-UTR sgRNA species. The 'binding site*' qPCR product is specific to the respective CRISPRi target position; 'nb' denotes a primer pair outside of the binding site that controls for nonspecific ChIP signal. Data from *n* = 3 biologically independent replicates are shown with the mean value indicated on top of the

bar. **c**, Top, western blot of cell extracts from cells treated with α-amanitin. Lanes: 1, untransfected cells (wild type, for endogenous α-amanitin-sensitive RPB1); 2, cells transfected with pWT; 3, with pC4; 4, with pC4 and CRISPRi *ETF1*. Molecular weight band positions are shown on the left (kDa). The membrane was cut in three pieces prior to probing with the antibodies and was put together before scanning. Bottom, RT–PCR with RNA from transfectants 2, 3 and 4. DNA ladder band positions are shown on the left (nucleotides). **d**, RT–qPCR *ETF1* APA assay (as in Fig. 6b) with RNA from transfectants 2–4, bearing α-amanitin (α-am)-resistant normal or slow Pol II (as in Fig. 7c), and samples 5 and 6, bearing endogenous wild-type (WT) Pol II and transfected with CRISPRi control or *ETF1*. Data from *n* = 2 biologically independent replicates are shown with the mean value indicated on top of the bar.

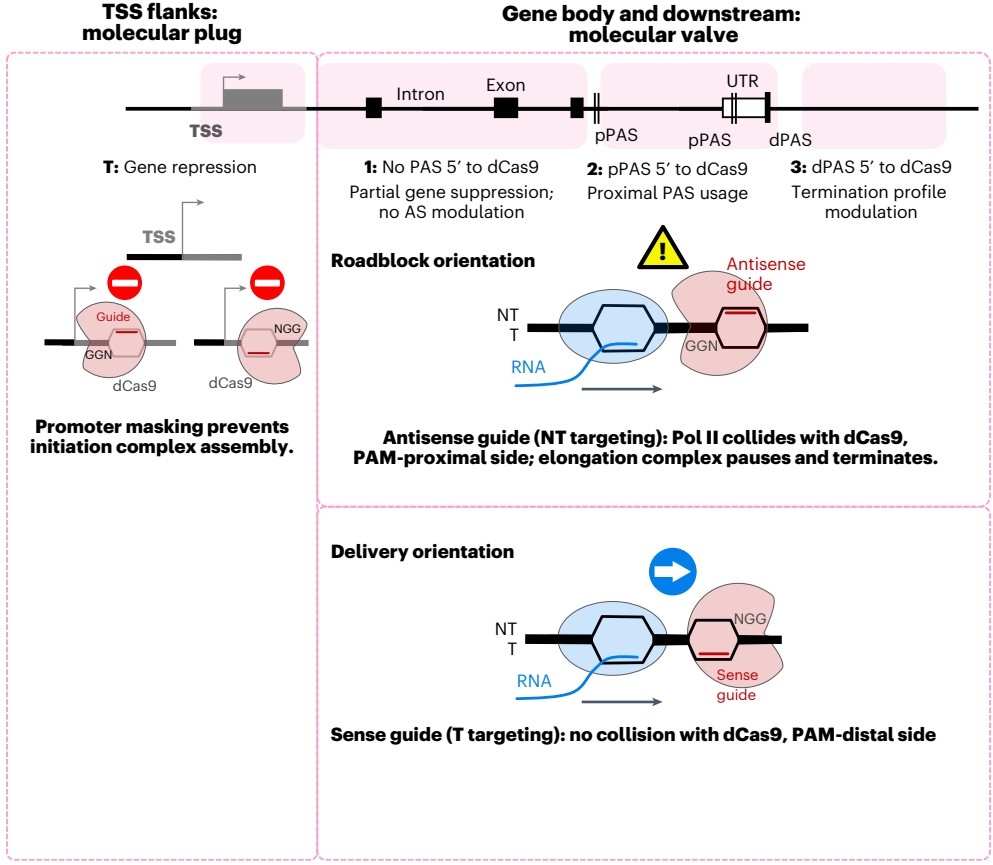

**Fig. 8 | dCas9 effects throughout the gene.** Diagram shows differential effects of dCas9 roadblocks when positioned in the TSS or regions 1–3 (targeting regions indicated in pink) in NT or T orientations. Proximal and distal PAS (pPAS, dPAS) are shown; other designations are as in Fig. 6a.

CRISPRi also stimulates the use of an intronic polyadenylation site in *PCMT1* (Fig. 6b), consistent with intronic PAS stimulation as observed previously in HEK293T-dCas9 cells for *RAD51C* and *ANKMY1* introns[37]. Interestingly, as with *TXNRD1*, CRISPRi downstream from the PAS in general did not suppress gene expression. Normalized mRNA levels upstream of the proximal PAS were unchanged for five of six genes, while RNA levels downstream of the PAS decreased, reflecting shifted PAS usage (Extended Data Fig. 5a). Only for *MRPS16* was alternative PAS stimulation accompanied by reduced gene expression, presumably due to altered mRNA isoform stability. Thus, we demonstrate that transient CRISPRi can indeed manipulate APA in endogenous genes without changing their nucleotide sequence.

We further analyzed RNA-seq data from cells transfected to express a mixture of sgRNA species targeting *ETF1*, *PCMT1*, *CCND1* and *H3-3B*. These CRISPRi APA cells have increased proximal polyadenylation of targeted genes, as clearly shown by the polyA+ RNA-seq profiles of *ETF1* and *H3-3B* (Fig. 6c). In agreement with our RT–qPCR data (Fig. 6b), the effect was less prominent but still evident for *CCND1* and *PCMT1* (Extended Data Fig. 5b). For *PCMT1*, CRISPRi APA mix data show a small but reproducible decrease in long-isoform terminal-exon read density, together with a reduction in spliced read count (Extended Data Fig. 5c). This is consistent with a shift to intronic polyadenylation over splicing and distal polyadenylation. DE analysis confirms that target gene expression is unaffected by CRISPRi APA (Extended Data Fig. 5d). Overall, our RNA-seq analysis of CRISPRi APA corroborates the above RT–qPCR analyses (Fig. 6a).

## AS is unaffected by CRISPRi

The APA analysis described above implies that restricted EC progression through the CRISPRi block can affect PAS usage. We therefore tested whether CRISPRi has a similar effect on AS, as this process is known to be influenced by changes in transcription elongation rate[1,39,40]. Specifically, slow elongation provides more time for either recognition of suboptimal 3′ splice sites by the spliceosome (type 1, included exons) or for binding of negative splicing factors (type 2, skipped exons). Moreover, the same AS changes can be stimulated by antisense small interfering RNA (siRNA)[41] or antisense chemically modified oligonucleotide[42] targeted downstream of an alternative exon. This antisense targeting triggers transcriptional gene silencing through heterochromatin formation in the neighboring DNA and ultimately reduces local Pol II processivity[41,42]. We reasoned that Pol II paused by CRISPRi is likely to slow down before the pausing, and this may affect AS similarly to antisense nucleic acid. Therefore, we applied CRISPRi to two elongation rate-dependent alternative exons[39], *TMEM188* (*CNEP1R1*) E2 (type 1) and *TBX3* E2A (type 2), using three sgRNA species targeting the NT strand for each gene. RNA from transfected cells was analyzed by endpoint PCR with reverse transcription (RT–PCR) to estimate the levels of mRNA with either skipped or included alternative exons. As a positive control, the DNA topoisomerase I inhibitor camptothecin (CPT) was employed, as it impedes EC progress genome wide and has previously been shown to influence AS[40,43]. As expected, CPT treatment stimulated *TMEM188* E2 inclusion and *TBX3* E2A skipping, respectively. However, CRISPRi targeting to these genes had no effect on AS events (Fig. 7a).

We next performed ChIP–qPCR analysis to evaluate dCas9-binding efficiency. For both for *TBX3* and *TMEM188*, we observe dCas9 binding and Pol II pausing over the targeted area (Fig. 7b) at levels similar to those observed for CRISPRi *TXNRD1* DS2-expressing cells. Interestingly, *ETF1* CRISPRi-induced Pol II pausing was less pronounced than for *TBX3* and *TMEM188*, but even so it still efficiently stimulated *ETF1*

proximal PAS usage (Fig. 6b). Thus, we demonstrate that, unlike APA, elongation rate-dependent AS is unaffected by targeted EC pausing. This suggests that dCas9 acts as an isolated EC roadblock, which is not accompanied by upstream transcriptional slowdown. Thus, the dCas9–sgRNA effect is very different than that of antisense siRNA, which results in a patch of slowly transcribed heterochromatin but does not induce abrupt pausing.

To further elucidate differences in stimulated APA and AS, we employed a slowly elongating Pol II mutant. Briefly, cells were transfected with the constructs encoding α-amanitin-resistant Pol II large subunit RPB1 supporting either normal (pWT) or slow (pC4) elongation rate[1]. Before analysis, transfected cells were treated with α-amanitin to induce degradation of endogenous RPB1 (Fig. 7c). Notably, western blot analysis with anti-Pol II NTD antibody showed that both pWT- and pC4-transfected cells display typical doublet ~240-kDa bands of Pol IIo (RPB1-phospho-CTD) and IIa (RPB1-CTD). They also revealed a 180-kDa band (IIb), corresponding to the truncated RPB1 isoform lacking the CTD[1].

As expected, cells with the slow Pol II mutant (pC4) showed increased *TMEM188* E2A exon inclusion and *TBX3* E2 skipping (Fig. 7c). However, with the same pC4-bearing cells, we did not detect an increase in *ETF1* short UTR isoform signal as compared to that of the pWT mutant (Fig. 7d). Similarly, the CRISPRi *ETF1* construct stimulated short UTR expression to the same extent in wild-type and pC4 cells, indicating that pC4 did not affect *ETF1* APA. These results indicate that, unlike targeted pausing, a general EC slowdown is insufficient to stimulate upstream PAS usage.

In sum, our data demonstrate an important difference between targeted CRISPRi elongation pausing (roadblock) and general elongation inhibition. While AS is a kinetically dependent process, APA requires targeted pausing followed by transcription termination. Apparent AS insensitivity to CRISPRi indicates that the EC does not slow down before encountering the block but instead pauses or collides with it. We demonstrate that dCas9 has an all-or-none effect, with a choice between either premature termination or transcription reading through the block.

## Discussion

This study analyzes the effects of CRISPRi targeting outside of the usually targeted promoter regions of human genes. Notably, dCas9 binding to endogenous gene targets affects gene expression in different ways, reflecting a localized transcriptional roadblock effect rather than a wider reduction in Pol II processivity (Fig. 8).

Even though dCas9 binds equally well to gene targets in both orientations, binding to sense target sites (with the PAM sequence facing the 3′ end of the gene) causes minimal transcription disturbance, while binding to antisense sites (PAM facing the 5′ end) creates a transcriptional obstacle (Fig. 3). Cryo-electron microscopy data[30] demonstrated that dCas9–sgRNA binding to the target site stabilizes the adjacent PAM-proximal DNA region, which is not in direct contact with the complex, ultimately creating a unidirectional roadblock to the Pol II EC. While this paper was under revision, the polarity of dCas9 roadblocks toward EC progression was independently shown in a bacterial in vitro system by single-molecule assays[32]. Altogether, the asymmetry of dCas9 effects suggests that, when CRISPR–dCas9 systems are used to study the effects of sequence-specific modifications or for imaging applications, the use of sense target sites is preferable. This will minimize confounding effects of dCas9 binding on transcription without sacrificing binding efficiency.

We argue that the dCas9 roadblock induces artificial pause-induced transcription termination via a torpedo mechanism. However, despite the previously reported association of pause-induced termination with H3K9me2 and H3K9me3, CRISPRi termination does not induce accumulation of these marks (Fig. 2). Thus, we suggest that dCas9 blocks elongation directly. Moreover, we demonstrate that a

local increase in H3K9 methylation is insufficient to induce transcription termination or pausing.

We demonstrate that dCas9 with an antisense guide presents a strong roadblock for the Pol II EC, irrespective of the target site location within the transcription unit. The presence of a sharp Pol II ChIP–qPCR peak upstream of the block (Fig. 4b) implies that Pol II–dCas9 collision results in a transient but quickly resolved pausing event rather than a long-term transcription arrest with secondary Pol II–Pol II collisions or a queue of ECs (which would presumably result in a wider ChIP–qPCR peak). Observed Pol II pausing can be followed by pause-induced transcription termination. If there is an active PAS upstream of the block, termination results in a shifted termination zone (Fig. 4b) but unchanged gene expression (Fig. 4c). Conversely, when the CRISPRi target is not in the vicinity of an active upstream PAS, the prematurely terminated transcript cannot be polyadenylated and is ultimately degraded with consequent gene suppression (Fig. 4c).

A fraction of Pol II ECs encountering dCas9, however, appear to complete the transcription cycle normally. Indeed, the Pol II *TXNRD1* 3′-end termination profile of CRISPRi in2-transfected cells (Fig. 4b) is identical in control cells. This outcome would either require Pol II transcription through the block or represent heterogeneity of dCas9 binding across the cell population, with dCas9 not binding or temporarily dissociating from the DNA in a subpopulation of transfected cells. Notably, we find minimal nonspecific genome-wide effects of CRISPRi targeting (Fig. 5b). However, we note that inhibition of upstream transcription can indirectly activate a downstream promoter (Fig. 5c), which we suggest might be a common side effect in CRISPRi knockdown screens.

We argue that APA stimulation is a variation of the induced termination scenario, arising when the target is located between two active PAS. In this scenario, induced termination in the inter-PAS area stimulates proximal PAS usage (Fig. 6a,b) and subsequent release of a shorter polyadenylated transcript. Strikingly, APA stimulation is only observed when Pol II transcription is blocked by CRISPRi but not when a slowly elongating Pol II mutant is used. This underlines the importance of pausing kinetics in cleavage and polyadenylation. At the same time, CRISPRi does not induce widespread transcription slowdown and so cannot modulate elongation rate-dependent AS events (Fig. 7a,b). Instead, it induces EC pausing or stalling exactly at the Pol II–dCas9 collision point. This collision-induced pausing correlates well with the study in bacteria, in which RNA-seq read density from immunoprecipitated RNA polymerase drops abruptly 19 bp upstream from the dCas9-binding site, in accordance with the distance from the polymerase active center to its front edge (Fig. 3 in ref. 12). Therefore, AS decisions are unaffected by CRISPRi blocks, as transcription stops so that the downstream transcript is never synthesized. Targeting the NT strand within gene bodies will generate an abortive transcript, inducing a decrease in gene expression. We suggest that, in all these scenarios, CRISPRi interaction with the Pol II EC is essentially the same. The ultimate gene expression outcome is defined by the target site context.

## Online content

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

## Methods

### Cell culture

All kits and reagents used in this section and the following sections are listed in Supplementary Table 1.

HeLa cells (originally obtained from ATCC (CCL-2) and maintained in the Proudfoot laboratory) and HCT116 XRN2-AID TIR1 cells (a gift from S. West's laboratory (Exeter University)[16]) were maintained in high-glucose DMEM medium with 10% FBS at 37 °C with 5% $CO_2$.

HeLa cells were plated at 130,000–150,000 in 2 ml of medium per well in a six-well plate (scaled up accordingly for larger vessels) the day before the experiment.

Transfections were performed with 1 μg DNA per 2 ml medium (typically, 1 μg main plasmid, 50–100 ng tracking pMax-GFP plasmid, 2 μl X-tremeGENE 9 reagent in 100 μl Opti-MEM transfection medium). The mixture was incubated at room temperature for 20 min and then added to the cells; 24 h later, medium was replaced with selective medium (2.5 μg ml$^{-1}$ puromycin), and, 24 h later, cells were collected for analysis.

HCT116 XRN2-AID cells were plated at 400,000 in 2 ml medium the day before transfection. The transfection mix contained 2 μg main plasmid, 50–100 ng tracking pMax-GFP plasmid and 8 μl X-tremeGENE HP reagent in 200 μl Opti-MEM. The mix was incubated at room temperature for 20 min and then added to the cells; 24 h later, medium was changed, and, 24 h later, cells were collected. To induce XRN2-AID depletion, 500 μM auxin (0.5 M in ethanol) was added 2 h before collection. For cell sorting, cells were trypsinized and resuspended in sorting buffer (PBS, 10% FBS), and GFP-positive cells were selected using the BD FACSAria III cell sorter (BD Biosciences) with FACSDiva software (BD Biosciences). After sorting, pelleted cells were used for RNA preparation as described below.

Other treatments were performed as follows:

- To induce osmotic stress, KCl (2 M in water) was added to a concentration of 80 mM 1 h before collection.
- UNC-0638, BIX-02194 and VPA were added at the indicated concentrations 48 h before collection (30 min before transfection), and then, 24 h before collection, medium was changed and drug was added again.
- CPT (4 mM in DMSO) was added at 4 μM, 4 h before collection.
- α-Amanitin (1 mg ml$^{-1}$ in water) was added at 20 μg ml$^{-1}$ 24 h before collection.

For control treatments, an equal volume of drug solvent was added.

### sgRNA design and cloning

All plasmids used in the study are listed in Supplementary Table 3 (refs. 44–46). Guide RNA species targeting the NT strand were designed using CRISPOR software[47]. Blocks of two to four guides were normally designed for one region, with distance between the guide target sites of at least 30 nucleotides. Forward and reverse oligonucleotides for cloning the guides were generated with CRISPOR[47] (http://crispor.tefor.net) with settings for 'U6 expression from an Addgene plasmid', 'pX330-U6-Chimeric_BB-CBh-hSpCas9 (Zhang lab) + derivatives' and then 'Primers for gN20 guide'. These settings add a 5′-G to guide sequences starting with A/T/C to optimize Pol III transcription. Specific target sequences are listed in Supplementary Table 4.

To insert the guide into the CRISPRi construct, oligonucleotides were phosphorylated, annealed and ligated with the pIZ60 or pIZ65 plasmid linearized with BbsI (BpiI), as described in the Zhang laboratory protocol (https://media.addgene.org/data/plasmids/62/62987/62987-attachment_GcKIw4gnwq57wq_Din8.pdf).

### Western blot analysis

Cells were washed and trypsinized, and the resulting pellet was lysed in RIPA buffer (50 mM Tris, pH 8.0, 150 mM NaCl, 1% NP-40, 0.1% SDS, 0.5% sodium deoxycholate; 50 μl for ~200,000 cells) and treated with 0.2 μl benzonase (250 U μl$^{-1}$) per 50 μl of lysate at 37 °C for 10–20 min. Protein concentration was determined using the Bio-Rad protein assay. Total protein lysate (5–20 μg) was used for the western blot. The electrophoresis was run at 30 mA per minigel or at 120–180 V per camera.

For Pol II blots, home-made 6% acrylamide gels (37.5:1 acrylamide:bis-acrylamide ratio) were prepared and run for 1 h at 30 mA per minigel and transferred to 0.4-μm nitrocellulose membranes in standard Laemmli TB for 2 h at 200 mA with an ice block.

For histone blots, premade 12% Bolt Tris–tricine gels (Novex) were run in proprietary Novex MES running buffer according to the manufacturer's manual and transferred to 0.2-μm nitrocellulose membranes in Novex transfer buffer for 1 h at 200 mA with an ice block.

After transfer, all membranes were blocked with 2% milk in TBS-T (50 mM Tris, pH 8.0, 150 mM NaCl, 0.5% Tween-20), incubated overnight with primary antibodies (Supplementary Table 2), washed three times with TBS-T, incubated for 30–60 min with secondary antibodies and developed using the Odyssey infrared scanner.

### RNA preparation and reverse transcription analysis

For total RNA preparation, TRI reagent (Sigma) was added directly to cells on the plate or cell pellets (500 μl for cells in 2 ml medium). After resuspending cells completely, 100 μl chloroform was added, the solution was mixed well by shaking, and tubes were centrifuged for 15 min at +4 °C and 16,000 rcf. The supernatant (~250 μl) was mixed with 200 μl isopropanol, incubated for 10 min at room temperature and centrifuged for 15 min as described above. The RNA pellet was washed twice with 80% ethanol and once with 96% ethanol, air dried and resuspended in 150 μl DNase mix (with 15 μl TURBO DNase buffer, 3 μl DNase TURBO (2 U μl$^{-1}$) and water) and incubated for 30 min at 37 °C. The resulting RNA was purified using an RNA clean and concentrator kit (Zymo Research) and used for the reverse transcription reaction. If negative control no-reverse transcription reactions demonstrated noticeable qPCR signal, DNase TURBO treatment of RNA was repeated, and a new reverse transcription reaction was set up.

For transcription-termination analysis, reverse transcription reactions were performed in 10 μl; briefly, 0.5–2 μg total RNA was premixed with 0.5 μl 10 mM dNTPs and 0.25 μl random primers (Invitrogen, 3 μg μl$^{-1}$) in 7 μl; the primer was annealed at 65 °C for 5 min, the reaction was cooled to 0 °C for 5 min, and 3 μl enzyme premix was added containing 0.25 μl SuperScript III for reverse transcription reactions and 0.25 μl water for no-reverse transcription reactions. Reactions were carried out for 10 min at 25 °C followed by 50 min at 50 °C and 5 min at 85 °C (enzyme inactivation). The resulting reaction mix, containing cDNA, was diluted tenfold or 100-fold depending on target abundance, and 2 μl was used per 18 μl qPCR reaction with the SensiMix qPCR mix, containing 0.3 μM forward and reverse primers, with qPCR performed on Rotor-Gene 3000 machines. Reactions were analyzed either by the ΔΔCt method (for spliced RNA) or against standard curves, prepared with serial dilutions of 300–500-bp fragments of HeLa genomic DNA.

For mRNA analysis, for a 10-μl reaction, 0.5–2 μg total RNA was premixed with 0.5 μl 10 mM dNTPs and 0.5 μl 50 μM phased oligo-dT primer ($T_{17}V$) and incubated at 65 °C for 5 min followed by incubation at 0 °C for 5 min. After that, 3 μl premix containing 0.5 μl SuperScript IV was added; the reaction was incubated at 50 °C for 10 min, 55 °C for 10 min and 80 °C for 10 min (enzyme inactivation). Resulting cDNA-containing reactions were diluted and used for qPCR as described above by the ΔΔCt method.

### Chromatin immunoprecipitation

ChIP was performed according to the CST Easy ChIP Enzymatic protocol (CST, 9003). Briefly, cells were taken out of the incubator and cross-linked for 10 min at room temperature with shaking, with 1% formaldehyde (Sigma, 37.5% molecular biology grade) added directly to the culture medium. Cross-linking was quenched with 0.125 M glycine

for 3 min with shaking. Plates were washed twice with ice-cold PBS, and cells were scraped and centrifuged at 2,000$g$ for 5 min in the cold room. The resulting pellets were snap frozen in liquid nitrogen and stored at −80 °C until use. Later, chromatin was prepared according to enzymatic ChIP kit (CST) instructions. Briefly, frozen pellets from single 10-cm plates were thawed for 10 min on ice, lysed with 2 ml buffer A (with protease and phosphatase inhibitors), centrifuged, resuspended in 2 ml buffer B, centrifuged, resuspended in 250 µl buffer B and treated with 1,000 U MNase for 20 min at 37 °C and 1,400 r.p.m. Enzymatic treatment was quenched with 25 µl 0.5 M EDTA, and samples were centrifuged and resuspended in 200 µl ChIP buffer with protease and phosphatase inhibitors. Nuclei were lysed on ice for 10 min and sonicated for five cycles (30 s on, 30 s off) on the medium setting of the Bioruptor sonicator. Tubes were centrifuged for 10 min at 9,400$g$ and +4 °C, and the supernatant was stored at +4 °C or −80 °C to be later used for ChIP. Concentration and fragment size of the resulting chromatin was checked by decross-linking and purifying a 15-µl aliquot of the resulting chromatin, followed by fractionation on a 1.5% agarose TAE gel (100 V for a 20-cm camera).

For ChIP, chromatin was diluted to 12 ng µl$^{-1}$ DNA, and normally 3 µg (250 µl) was used for ChIP (Supplementary Table 5). Immunoprecipitation was performed overnight in the cold room on a rotating wheel, and then magnetic beads were added for an additional 2 h. Beads were washed three times for 5 min on a wheel with low-salt buffer (20 mM Tris, pH 8, 150 mM NaCl, 2 mM EDTA, 0.5% NP-40, 0.1% SDS) and once for 5 min with high-salt buffer (same buffer but with 500 mM NaCl). All supernatant was aspirated completely, and 100 µl decross-linking mix (93 µl Milli-Q water, 5 µl 5 M NaCl, 1 µl 20 mg/ml Proteinase K) was added. Samples were decross-linked for 2–4 h at 65 °C with shaking, and DNA was purified from the supernatant using a ChIP DNA clean and concentrator kit. The resulting DNA was diluted (normally to 25–50 ng µl$^{-1}$ DNA, that is, IP with 3,000 ng was diluted to 120 µl), and 2 µl was used for qPCR with SensiMix as described for RNA analysis and analyzed using the ΔΔCt method versus input samples.

Enrichment is either presented raw (for dCas9 IP) or normalized as described in figure legends. For instance, for Fig. 2b, signal was normalized to the maximal signal downstream from the PAS to make termination easily comparable between experiments.

## Chromatin and polyA+ RNA sequencing

HeLa cells were transfected in 15-cm plates as described above with *TXNRD1* gene-specific gRNA species alone or a mix of gene-specific gRNA species for APA experiments. A quarter of the transfected cells were used to extract total RNA (TRIzol); 0.5 µg of total RNA was used to prepare polyA+ RNA libraries using the NEBNext Ultra II Directional RNA Library Prep Kit with the NEBNext Poly(A) mRNA Magnetic Isolation Module. Chromatin RNA was extracted from the other three quarters of transfected cells as described previously[17] with the following modifications. The chromatin pellet was digested with 2 µl TURBO DNase (Life Technologies) in 200 µl high-salt buffer (10 mM Tris-HCl, pH 7.5, 500 mM NaCl and 10 mM MgCl$_2$) for 15 min at 37 °C and then treated with proteinase K in 0.2% SDS for 10 min at 37 °C. Chromatin RNA was extracted with the phenol–chloroform method, and a second round of TURBO DNase digestion was performed, followed by RNA extraction using TRIzol. One microgram of chromatin RNA was ribodepleted using the RiboCop rRNA Depletion Kit (Lexogen), followed by library preparation using the NEBNExt Ultra II Directional RNA kit. All libraries were sequenced on the NovaSeq 6000 by Novogene UK.

## Splicing analysis

RNA was extracted as described above, and splicing analysis was performed as previously described[39]. Briefly, 1 µg of total RNA was reverse transcribed using SuperScript III (Thermo Fisher Scientific) reverse transcriptase and 100 ng of random primers. GoTaq (Promega) was used for PCR amplification (30 cycles for *TMEM188* E2 and 27 cycles for *TBX3* E2A), with 1.5 mM MgCl$_2$ and specific gene primers at 0.3 µM each (see Supplementary Table 5 for primer sequences). Products were loaded on a 6% acrylamide (37.5:1) TBE gel and stained with ethidium bromide for visualization.

## Computational analysis

**Preprocessing of Illumina reads.** PolyA and chromatin paired-end strand-specific RNA-seq raw Illumina short reads in FASTQ format were quality controlled with FastQC (https://www.bioinformatics.babraham.ac.uk/projects/fastqc/), without any unexpected bias found. Trim Galore (https://www.bioinformatics.babraham.ac.uk/projects/trim_galore/) was used in paired-end mode to trim read adaptors as well as to remove reads with low-quality ends (Phred score cutoff of 20) and/or less than ten nucleotides. Selected reads were aligned against the reference human genome (GRCh38) using STAR software[48], requiring a minimum alignment score (–outFilterScoreMin) of 10 and excluding non-uniquely mapped reads (–outFilterMultimapNmax 1).

**Gene profiles.** Uniquely mapped reads contained in BAM files were divided into forward and reverse strands, according to their bitwise flags 83 163 and 99 147, respectively. SAMtools[49] was used to perform this task. Next, strand-specific BAM files were converted into bedGraph format with 'bedtools genomecov'[50], and 'bedGraphToBigWig' was used to compress them into bigwig files. Each bigwig was normalized to the library size (number of paired-end fragments) aligned in the original BAM file. The $y$ scale represents read counts per million mapped reads. Bigwig files were visualized with the UCSC Genome Browser and exported in a PDF.

**Reference gene annotations.** All transcriptional units considered in the downstream analysis were based on the Ensembl human (hg38) reference gene annotation, version 108.

**Differentially expressed genes.** Kallisto[51] was used to map polyA RNA-seq reads against the human transcriptome ('cdna' and 'ncRNA' FASTA files) to produce estimated gene expression values, which were then gathered in a non-normalized count matrix. Significant differentially expressed genes were detected with the DESeq2 package[52] by using the created matrix as input. To remove noise while preserving large differences, the 'lfcShrink'[53] function with argument 'type = 'apeglm'' was used to extract results from DESeq2. Cutoffs of $1 \times 10^{-5}$ for $P$ values and 0.5 for the absolute value of log$_2$ (fold change) were applied over DESeq2's own two-sided statistical test results. Volcano plots were generated with the ggplot2 package[54].

Plots showing expression of individual genes were generated with the 'plotCounts' function from the DEseq2 package. Read counts were normalized with scaled factors discovered by DESeq2 internally using the median ratio method.

**Heatmaps comparing chromatin RNA-seq signal along genes with different sgRNA species.** To enable comparison between loci of different samples, size factors were estimated with the 'estimateSizeFactors' function, part of the DESeq2 package, on chromatin RNA-seq samples. As input, we used raw counts per gene obtained with featureCounts[55] grouped at the gene level (-g gene_id).

Next, the *TXNRD1* gene locus (chromosome 12:104,286,383–104,372,549) was divided into adjacent bins of 500 bp. The number of mapped fragments overlapping each bin was obtained using the pysam package[49] in Python and then divided by the previously calculated size factors. The plotted values in the heatmap are the log$_2$ (fold change) of signal found in the sgRNA-treated sample versus the control sample. $P$ values were calculated based on the expectation found in 1 million simulations of a binomial distribution with a probability of 0.5 and $n$ equal to the sum of normalized reads found in the two samples that

were being compared. Bins with $P$ value less than $1 \times 10^{-5}$ were kept. Final heatmap plots were created with ggplot2 in R.

Using the *TXNRD1* gene as a reference, a set of untargeted genes with similar length (±2,500 bp) and expression (±5 transcript per million – TPM) were used as negative controls. The methodology applied to them was the same as that applied for *TXNRD1*.

**Sashimi.** Sashimi plots were created with ggsashimi[56] on polyA RNA-seq samples. The following parameters were used: '-M 80 -s MATE2_SENSE -S plus'. The number present in each splice junction represents the number of reads found in the BAM file supporting its existence. In sashimi plots with one track per condition only, the number of read counts supporting each event was internally aggregated among the three replicates using the arithmetic mean.

### Statistics and reproducibility
No statistical methods were used to predetermine sample size; sample size was taken as the number of biological replicates. No data were excluded from analyses; all experiments and assays were confirmed with at least one replicate as described in the text. Experiments were not randomized. All cell cultures were grown under identical conditions; therefore, randomization was not relevant for this study. The investigators were not blinded to allocation during experiments and outcome assessment. Blinding is not applicable for this study, as it does not involve any subject assessment of the data that may influence the validity of the results.

### Reporting summary
Further information on research design is available in the Nature Portfolio Reporting Summary linked to this article.

### Data availability
All data needed to evaluate the conclusions in the paper are present in the paper. Genome-wide datasets are deposited at the GEO under the accession number GSE228798. All data and materials are available from the corresponding author upon request. Source data are provided with this paper.

### Code availability
No new code was generated for data analysis in this paper; all software and algorithms used for analysis are listed in Supplementary Table 6.

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

## Acknowledgements
We thank Y. Kainov, D. Ghilarov, I. Cowell, C. Austin, S. West, A. Kornblihtt and N.J.P. group members for critical discussions. We are also grateful to S. West (Exeter University) for providing HCT116 XRN2-AID cells, J. Riepsaame for advice and reagents for CRISPRi system setup and R. Hedley for cell sorting. This project was supported by funding from the European Union's Horizon 2020 research and innovation program under the Marie Skłodowska-Curie grant (agreement no. 747391) to I.Z. and a Wellcome Trust Investigator Award (107928/Z/15/Z) to N.J.P.

## Author contributions
I.Z. conducted all experiments except for splicing analyses and sample preparation for next-generation sequencing, performed by G.D. I.Z. and N.J.P. designed the project and co-wrote the manuscript with help from G.D. R.S.-L. performed next-generation sequencing computational analyses.

## Competing interests
The authors declare no competing interests.

## Additional information
**Extended data** is available for this paper at https://doi.org/10.1038/s41594-023-01090-9.

**Correspondence and requests for materials** should be addressed to Inna Zukher or Nick J. Proudfoot.

**a**

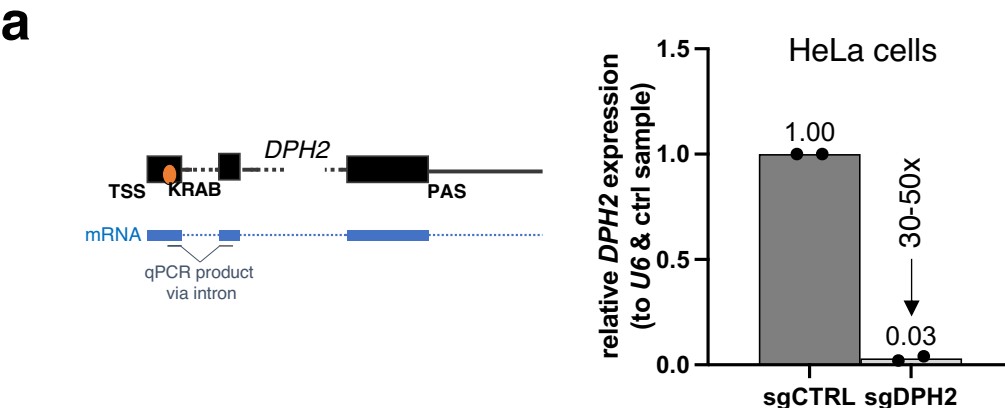

**b**

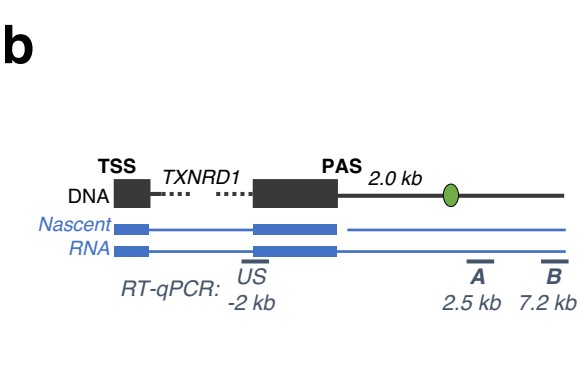

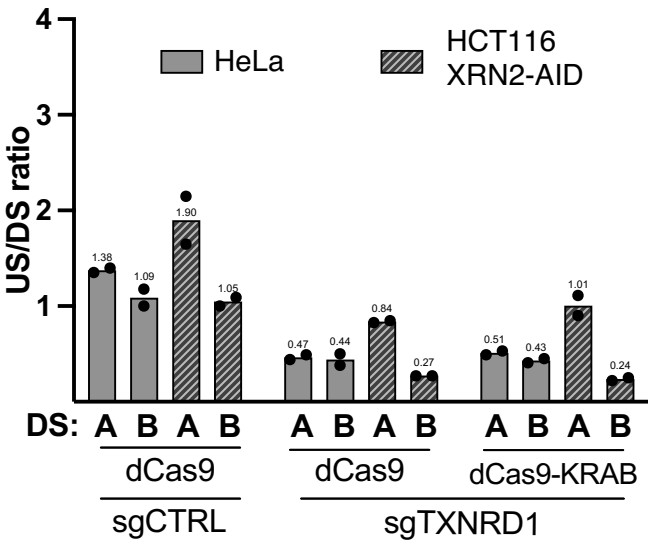

**Extended Data Fig. 1 | Validation of CRISPRi targeting TSS and PAS downstream gene regions.** a, CRISPRi-KRAB transcription repression. HeLa cells transfected with CRISPRi-KRAB constructs expressing non-targeting (CTRL) or gene-specific (*DPH2* TSS) sgRNAs for 48 h and extracted RNA analysed by RT-qPCR. Data from n = 2 biologically independent replicates are shown with mean value indicated on top of the bar. mRNA levels expressed as a normalized to DPH2/U6 expression level in (CTRL) sample. b, Transcriptional readthrough in HeLa vs XRN2-AID cells. HeLa or HCT116 TIR1 XRN2-AID (XRN2-AID) cells were transfected with CRISPRi or CRISPRi-KRAB, encoding non-targeting or *TXNRD1* sgRNAs. After 48h RNA was extracted from puromycin-resistant (HeLa) or GFP-positive (XRN2-AID) cell populations. Steady-state levels of downstream *TXNRD1* RT-qPCR probes (2.5 kb (A) and 7.2 kb (B) downstream form PAS) normalized to US probe (unspliced RNA) are shown. Data from n = 2 biologically independent replicates are shown with mean value indicated on top of the bar.

**a**

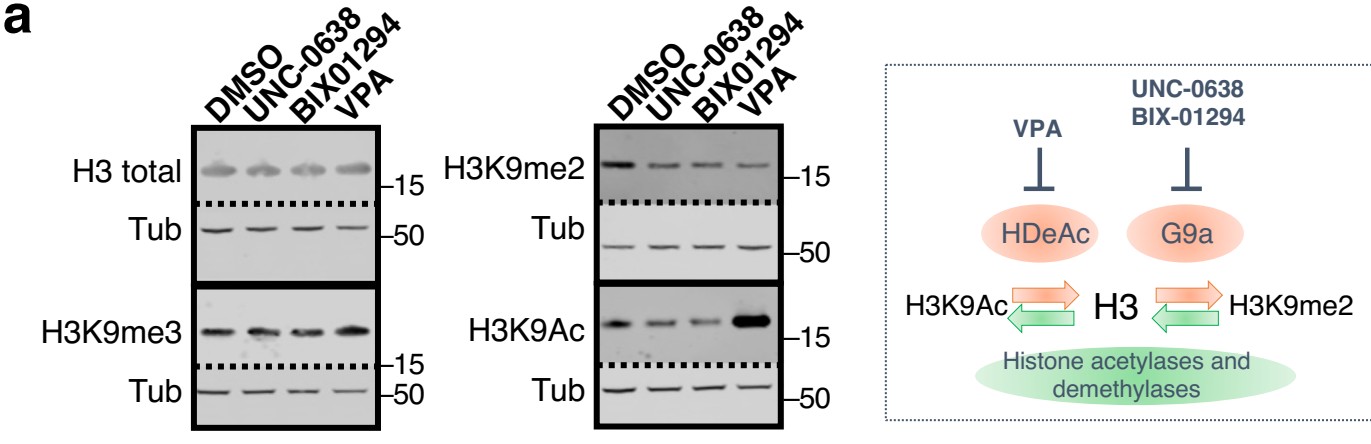

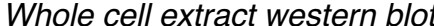

*Whole cell extract western blot*

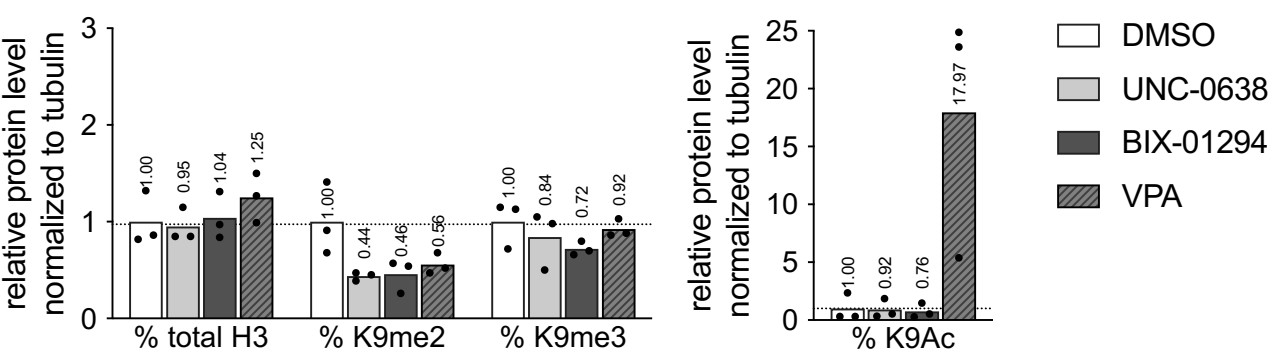

**b**

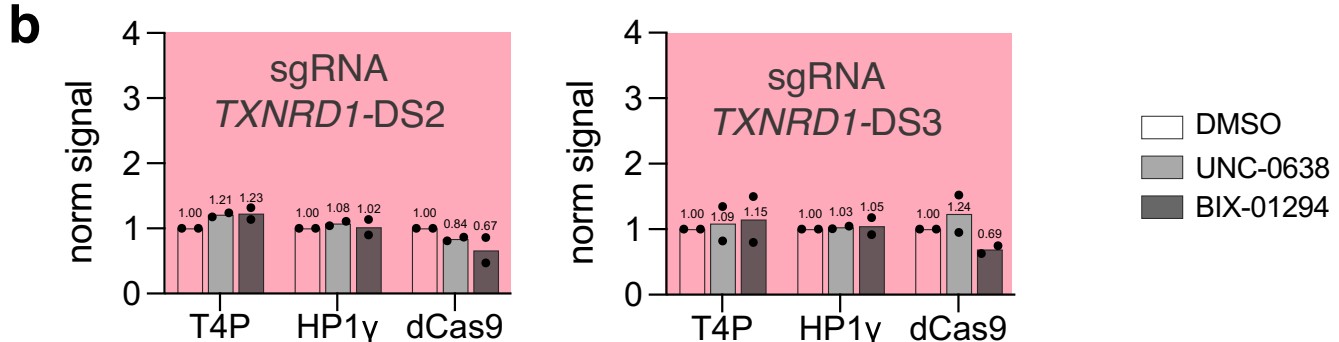

*ChIP-qPCR signal at dCas9 target site*

**Extended Data Fig. 2 | Repressive chromatin marks unaffected by CRISPRi.**
**a**, Representative western blot of protein extracts from HeLa cells, treated with G9a inhibitors (UNC-0638 or BIX-01294) or histone deacetylase inhibitor (VPA) as follows: 0.4% DMSO (mock sample), 250 nM UNC-0638, or 4 µM BIX-01294, or 10 µM VPA (valproic acid) for 48h prior to western blot. Molecular weight bands position shown on the right (kDa). Barcharts below show quantitation of western blot data from biologically independent replicates with mean value indicated on

top of the bar. Diagram shows histone methylase G9a and histone deacetylase (HDeAc) inhibitors action. **b**, HeLa cells were transfected with CRISPRi plasmids expressing *TXNRD1* DS2 or DS3 sgRNA sets and treated as in **a** prior to ChIP-qPCR. For each transfection ChIP enrichment signal is normalized to DMSO-treated sample (T4P and HP1γ samples additionally normalized to *MYC* 3′-end DNA to internally control IP efficiency). Data from n = 2 biologically independent replicates are shown with mean value indicated on top of the bar.

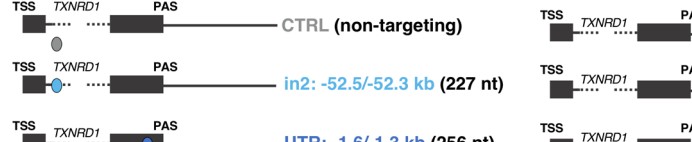

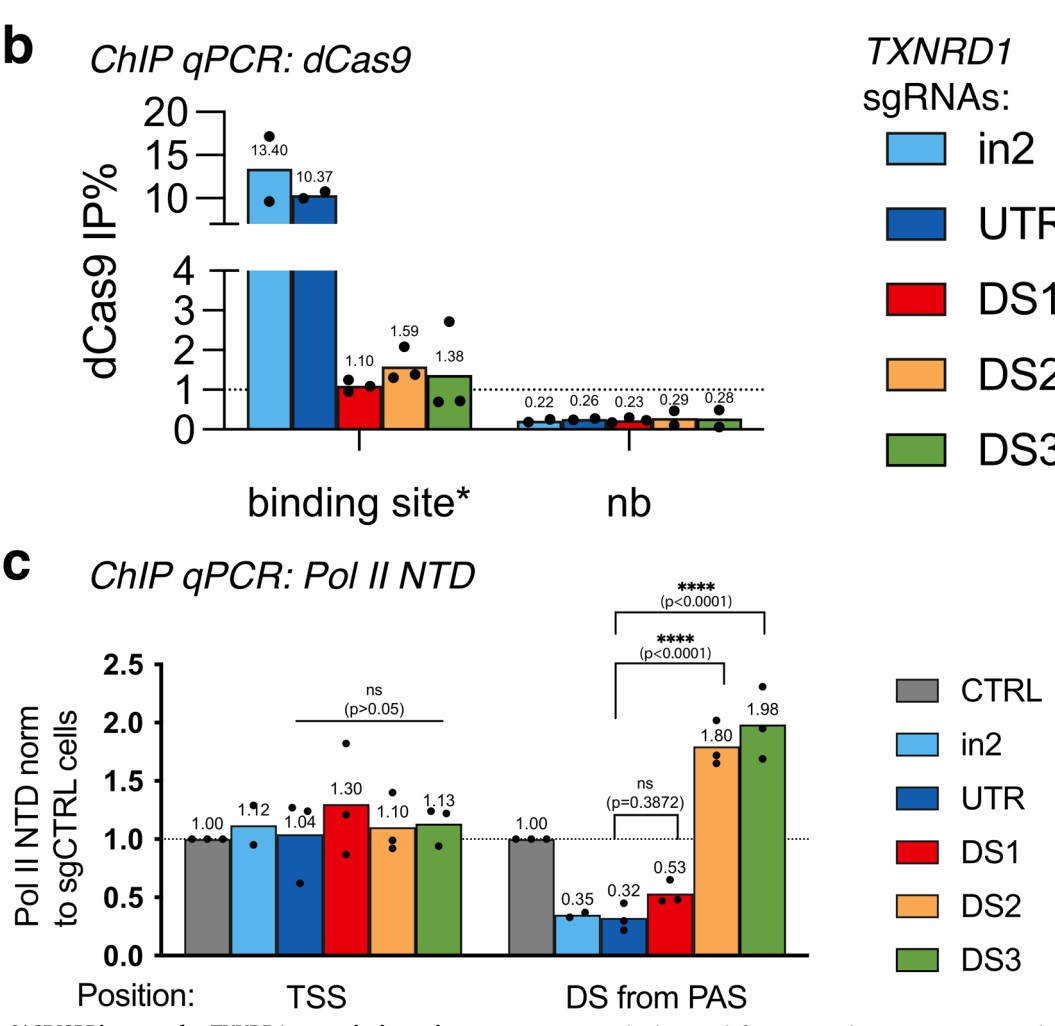

**Extended Data Fig. 3 | CRISPRi targeted to TXNRD1 transcription unit.**
**a**, Scheme depicting dCas9 targets throughout *TXNRD1*. **b**, ChIP-qPCR from cells as indicated with anti-FLAG antibody, recognizing FLAG tag on dCas9. Percent enrichment over input is shown, designated as in Fig. 3b. Data from individual biologically independent replicates are shown with mean value indicated on top of the bar. **c**, HeLa cells transfected as above prior to ChIP-qPCR with Pol II

NTD antibody. Signals from TSS or downstream qPCR products ('DS from PAS', includes PAS+1.2, 1.7 and 5.2 kb products) are summed and normalized to signal in control sample. Data from individual biologically independent replicates are shown with mean value indicated on top of the bar. For experiments with n≥3 one-way ANOVA with Tukey's multiple comparisons test shows difference as indicated.

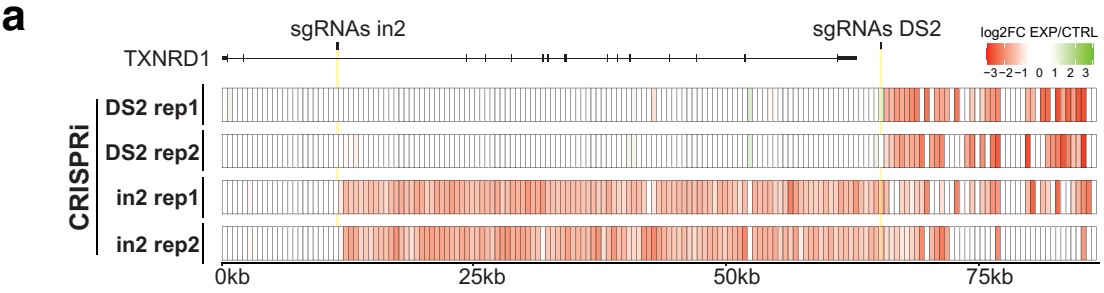

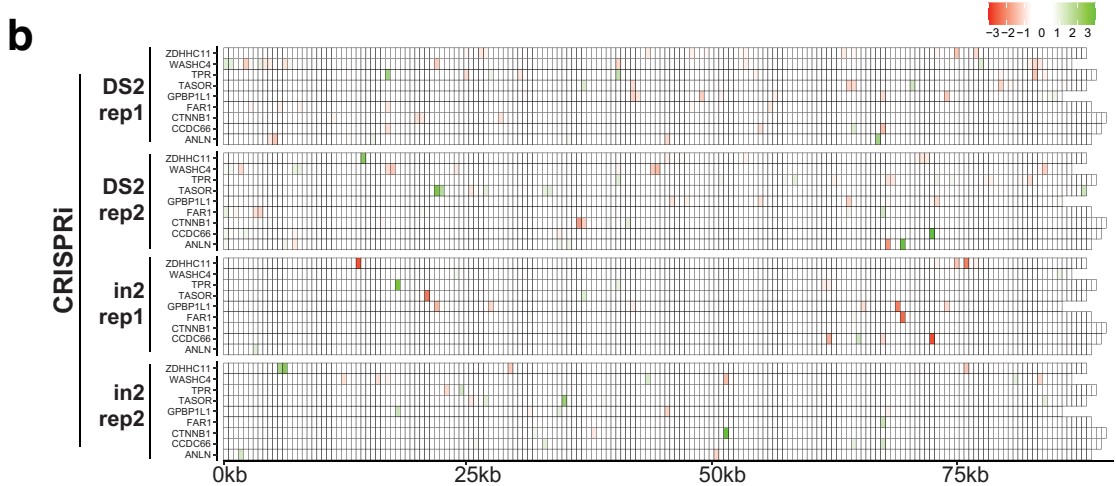

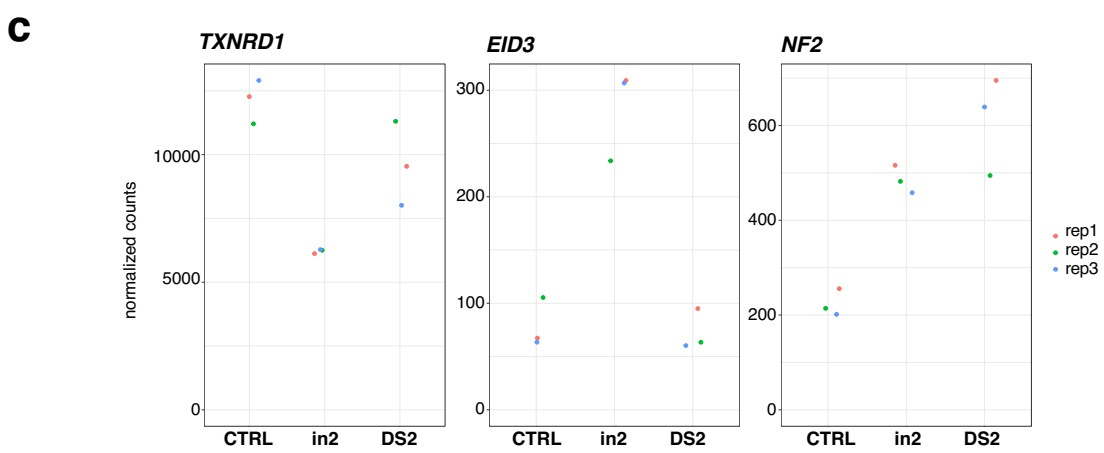

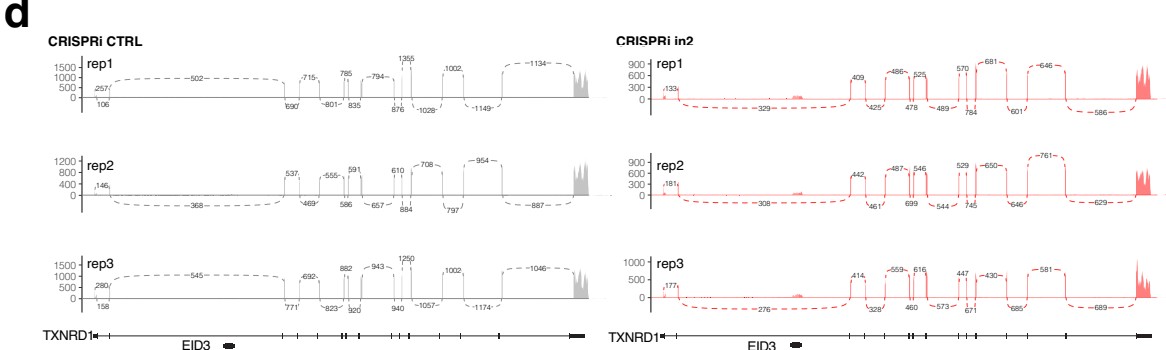

**Extended Data Fig. 4 | See next page for caption.**

**Extended Data Fig. 4 | Bioinformatic analysis of CRISPRi effects on TXNRD1.**
**a**, Binning analysis in the individual replicates (n = 2 bioiligically independent Chr RNA libraries; 500 bp per bin), same as in Fig. 5a. **b**, Binning analysis controls (nine genes similar to *TXNRD1* in length and expression (by transcript per million reads) selected and analysed as in Extended Data Fig. 4a). **c**, Normalized counts per gene for *TXNRD1*, *EID3* and *NF2* in polyA+ libraries. **d**, *TXNRD1* splicing sashimi plots for individual polyA+ replicates (as in Fig. 5c).

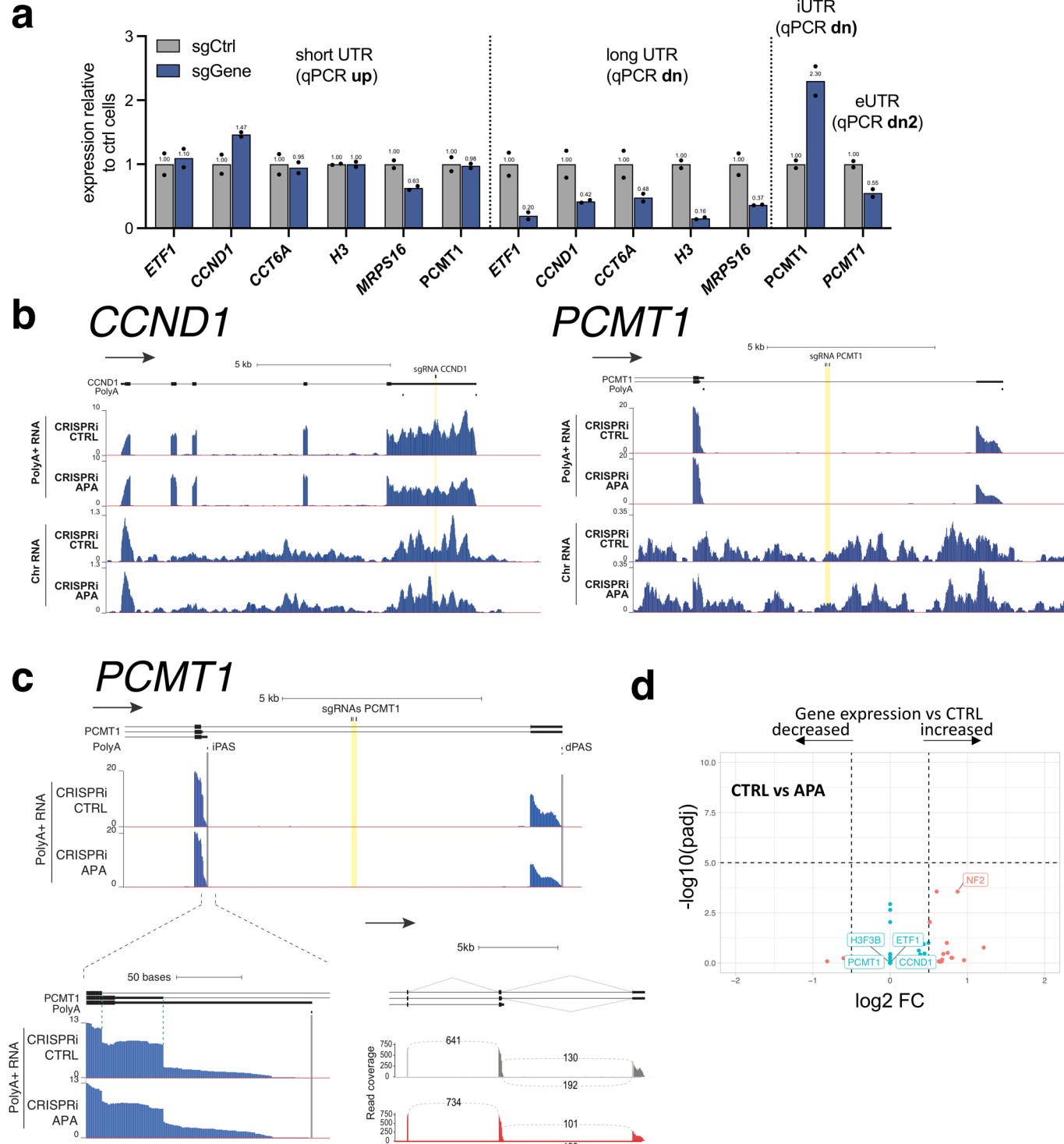

**Extended Data Fig. 5 | CRISPRi effects on APA. a**, CRISPRi targeted upstream of the terminal poly(A) site and gene expression level. RNA samples from Fig. 6a were reanalysed to estimate respective mRNA expression levels relative to control *MYC* gene expression and to upstream or downstream probe signal in the CTRL sample. Data from n = 2 biologically independent replicates are shown with mean value indicated on top of the bar. **b**, Screenshots from UCSC browser show representative Chr and PolyA+ RNA-seq profiles of HeLa cells transfected by CRISPRi CTRL or APA. *CCND1* and *PCMT1*. **c**, Zoom-in of *PCMT1* alternative PAS region. Dotted vertical lines show splice sites positions. Sashimi plot as in Extended Data Fig. 4d. **d**, Volcano plot of PolyA+ RNA-seq DE analysis (n = 3 biologically independent polyA+ libraries).

# Reporting Summary

## Statistics

For all statistical analyses, confirm that the following items are present in the figure legend, table legend, main text, or Methods section.

| n/a | Confirmed | |
|---|---|---|
| ☐ | ☒ | The exact sample size (*n*) for each experimental group/condition, given as a discrete number and unit of measurement |
| ☐ | ☒ | A statement on whether measurements were taken from distinct samples or whether the same sample was measured repeatedly |
| ☐ | ☒ | The statistical test(s) used AND whether they are one- or two-sided *Only common tests should be described solely by name; describe more complex techniques in the Methods section.* |
| ☒ | ☐ | A description of all covariates tested |
| ☒ | ☐ | A description of any assumptions or corrections, such as tests of normality and adjustment for multiple comparisons |
| ☐ | ☒ | A full description of the statistical parameters including central tendency (e.g. means) or other basic estimates (e.g. regression coefficient) AND variation (e.g. standard deviation) or associated estimates of uncertainty (e.g. confidence intervals) |
| ☐ | ☒ | For null hypothesis testing, the test statistic (e.g. *F*, *t*, *r*) with confidence intervals, effect sizes, degrees of freedom and *P* value noted *Give P values as exact values whenever suitable.* |
| ☒ | ☐ | For Bayesian analysis, information on the choice of priors and Markov chain Monte Carlo settings |
| ☒ | ☐ | For hierarchical and complex designs, identification of the appropriate level for tests and full reporting of outcomes |
| ☒ | ☐ | Estimates of effect sizes (e.g. Cohen's *d*, Pearson's *r*), indicating how they were calculated |

*Our web collection on statistics for biologists contains articles on many of the points above.*

## Software and code

Policy information about availability of computer code

| Data collection | No software was used for data collection. |
|---|---|
| Data analysis | FastQC (v0.11.5) https://www.bioinformatics.babraham.ac.uk/projects/fastqc/ TrimGalore (v0.6.7) https://www.bioinformatics.babraham.ac.uk/projects/trim_galore/ Samtools (v1.11)12 http://samtools.sourceforge.net/ [12] STAR (v2.7.0)13 https://github.com/alexdobin/STAR Kallisto (v0.46.0)14 https://github.com/pachterlab/kallisto [14] ggsashimi (v1.0.0)15 https://github.com/guigolab/ggsashimi [15] pysam (v0.15.4)12 https://github.com/pysam-developers/pysam [12] DESeq2 (v1.28.1)16 https://bioconductor.org/packages/release/bioc/html/DESeq2.html [16] lfcShrink (1.20.0)17 http://bioconductor.org/packages/release/bioc/html/apeglm.html [17] ggplot2 (v3.3.3)18 https://ggplot2.tidyverse.org Bedtools (v2.30.0)19 |

https://bedtools.readthedocs.io/en/latest/content/bedtools-suite.html
bedGraphToBigWig http://genome.cse.ucsc.edu/index.html
UCSC genome browser http://genome.cse.ucsc.edu/index.html
R (v4.2.1) https://www.r-project.org
Python (v3.10.6) https://www.python.org

For manuscripts utilizing custom algorithms or software that are central to the research but not yet described in published literature, software must be made available to editors and reviewers. We strongly encourage code deposition in a community repository (e.g. GitHub). See the Nature Portfolio guidelines for submitting code & software for further information.

## Data

Policy information about availability of data

All manuscripts must include a data availability statement. This statement should provide the following information, where applicable:

- Accession codes, unique identifiers, or web links for publicly available datasets
- A description of any restrictions on data availability
- For clinical datasets or third party data, please ensure that the statement adheres to our policy

All data needed to evaluate the conclusions in the paper are present in the paper. The NGS data were uploaded to the Genome Expression Omnibus (GSE228798 (https://www.ncbi.nlm.nih.gov/geo/query/acc.cgi?acc=GSE228798 )
Note: the data are currently in private mode, accessible to view with reviewer token (see below). Once the manuscript is accepted, the data will go public.
To review GEO accession GSE228798:
Go to https://www.ncbi.nlm.nih.gov/geo/query/acc.cgi?acc=GSE228798
Enter token orapcqqqrvydtqn into the box

## Human research participants

Policy information about studies involving human research participants and Sex and Gender in Research.

| | |
|---|---|
| Reporting on sex and gender | N/A |
| Population characteristics | N/A |
| Recruitment | N/A |
| Ethics oversight | N/A |

Note that full information on the approval of the study protocol must also be provided in the manuscript.

# Field-specific reporting

Please select the one below that is the best fit for your research. If you are not sure, read the appropriate sections before making your selection.

☒ Life sciences          ☐ Behavioural & social sciences          ☐ Ecological, evolutionary & environmental sciences

For a reference copy of the document with all sections, see nature.com/documents/nr-reporting-summary-flat.pdf

# Life sciences study design

All studies must disclose on these points even when the disclosure is negative.

| | |
|---|---|
| Sample size | No statistical methods were used to pre-determine the sample size. Sample size was taken as number of biological replicates. |
| Data exclusions | No data were excluded. |
| Replication | All experiments and assays were confirmed with at least one replicate as described in the manuscript. |
| Randomization | All cell cultures were grown under identical conditions therefore randomization was not relevant for this study |
| Blinding | Blinding is not applicable for this study as it does not involve any subject assessment of the data that may influence validity of results. |

# Reporting for specific materials, systems and methods

We require information from authors about some types of materials, experimental systems and methods used in many studies. Here, indicate whether each material, system or method listed is relevant to your study. If you are not sure if a list item applies to your research, read the appropriate section before selecting a response.

## Materials & experimental systems

| n/a | Involved in the study |
|-----|----------------------|
| ☐ | ☒ Antibodies |
| ☐ | ☒ Eukaryotic cell lines |
| ☒ | ☐ Palaeontology and archaeology |
| ☒ | ☐ Animals and other organisms |
| ☒ | ☐ Clinical data |
| ☒ | ☐ Dual use research of concern |

## Methods

| n/a | Involved in the study |
|-----|----------------------|
| ☒ | ☐ ChIP-seq |
| ☒ | ☐ Flow cytometry |
| ☒ | ☐ MRI-based neuroimaging |

## Antibodies

| Antibodies used | (Protein short name- Protein name - host animal - manufacturer - cat. no. - WB dilution and/or ChIP dilution)<br>Tub Tubulin Mouse Sigma T5168 1:10 000 –<br>FLAG anti-FLAG, M2, ChIP grade Mouse Sigma F3165 – 1 µg per 3 µg DNA<br>Pol II NTD ChIP grade Rpb1-NTD Rabbit CST #14958S 1:1 000 5 µl per 5 µg DNA<br>Pol II T4P ChIP grade Rpb1-CTD thr4P Rabbit CST #26319S – 5 µl per 5 µg DNA<br>HP1 γ ChIPAb+ validated HP1 gamma Mouse Sigma 17-646 – 5 µl per 3 µg DNA<br>H3 histone H3, ChIP grade Rabbit abcam ab1791 1:30 000 –<br>H3K9me2 H3K9me2 ChIP grade Mouse Abcam ab1220 1: 1 000 –<br>H3K9me3 H3K9me3, ChIP grade Rabbit abcam ab8898   1: 1 000 0.6 µg per 5 µg DNA<br>H3K9-ac H3K9 Acetyl, ChIP grade Rabbit abcam ab4441 1: 1 000 –<br> IRDye 800CW Goat anti-Mouse IgG Secondary Antibody Goat Licor P926-32210 1:10 000 –<br> IRDye 680RD Goat anti-Rabbit IgG Secondary Antibody Goat Licor P926-68071 1:10 000 – |
| Validation | All antibodies were obtained commercially and had been validated by the corresponding manufacturer |

## Eukaryotic cell lines

Policy information about cell lines and Sex and Gender in Research

| Cell line source(s) | Please refer to the Methods section. HeLa cells (originally obtained from ATCC (CCL-2) and maintained in Proudfoot lab) and HCT116 XRN2-AID TIR1 (Gift from Steve West lab, (Eaton et al., 2018)) |
| Authentication | not authenticated |
| Mycoplasma contamination | Cell lines tested negative for Mycoplasma contamination |
| Commonly misidentified lines<br>(See ICLAC register) | No misidentified cell lines were used in this study. |

