## [Peer Review File · Nature Structural & Molecular Biology]

Peer Review Information

Manuscript Title: Elongation roadblocks mediated by dCas9 across human genes modulate transcription and nascent RNA processing

Corresponding author name(s): Nick Proudfoot, Inna Zukher

Reviewer Comments & Decisions:

Decision Letter, initial version:
--

Message: Dear Dr Proudfoot,

Thank you again for submitting your manuscript "Elongation blockage by dCas9 modulates transcription and RNA-processing of human protein coding genes". I apologize for the delay in responding, which resulted from the difficulty in obtaining suitable referee reports. Nevertheless, we now have comments (below) from the 2 reviewers who evaluated your paper. In light of those reports, we remain interested in your study and would like to see your response to the comments of the referees, in the form of a revised manuscript.

You will see that all reviewers appreciate the results, some concerns regarding the strength of the data are raised by the Reviewer #1, who suggests that additional genomic analysis would make this a stronger candidate for NSMB. We agree with the referee that these additional experiments would make for a stronger study. Please also be sure to address all other technical comments.

We appreciate the requested revisions are extensive. We thus expect to see your revised manuscript within 6 months. If you cannot send it within this time, please let us know. We will be happy to consider your revision as long as nothing similar has been accepted for publication at NSMB or published elsewhere. Should your manuscript be substantially delayed without notifying us in advance and your article is eventually published, the received date would be that of the revised, not the original, version.

Reporting Summary:

Please note that all key data shown in the main figures as cropped gels or blots should be presented in uncropped form, with molecular weight markers. These data can be aggregated into a single supplementary figure. While these data can be displayed in a relatively informal style, they must refer back to the relevant figures. These data should be submitted with the last revision, prior to acceptance, but you may want to start putting it together at this point.

We require deposition of coordinates (and, in the case of crystal structures, structure factors) into the Protein Data Bank with the designation of immediate release upon

publication (HPUB). Electron microscopy-derived density maps and coordinate data must be deposited in EMDB and released upon publication. Deposition and immediate release of NMR chemical shift assignments are highly encouraged. Deposition of deep sequencing and microarray data is mandatory, and the datasets must be released prior to or upon publication. To avoid delays in publication, dataset accession numbers must be supplied with the final accepted manuscript and appropriate release dates must be indicated at the galley proof stage. Please find the complete NRG policies on data availability at <http://www.nature.com/authors/policies/availability.html>.

[Redacted]

Sincerely,

Carolina

Carolina Perdigoto, PhD
Chief Editor
Nature Structural & Molecular Biology
orcid.org/0000-0002-5783-7106

Referee expertise:

Referee #1: transcription regulation, CRISPR systems, genomics

Referee #2: transcription regulation

Reviewers' Comments:

Reviewer #1:

Remarks to the Author:

"Elongation blockage by dCas9 modulates transcription and RNA-processing of human protein coding genes" by Zukher, Dujardin, and Proudfoot investigate the utility and mechanism of catalytically inactivated Cas9 (dCas9) for prematurely blocking elongating PolII transcription complexes at various regions of gene bodies. They convincingly show that the orientation of dCas9 (targeting the template vs non-template DNA strand) within the transcription unit dictates whether or not blocking will occur. They go on to claim that dCas9 positioning within gene bodies, after gene bodies, or just downstream of alternative polyA sites can lead to mRNA suppression, readthrough suppression, or alternative polyadenylation. They attempt to similarly alter splice isoforms with a similar strategy, which proves unsuccessful. They also attempt to address mechanism of premature elongation termination beyond a simple roadblock model by examining factors such as XRN2-mediated termination, and a potential role of HP1gamma and H3K9methylation.

In summary, I think the study is interesting and useful for others working with CRISPRi or alternative polyadenylation. However, some claims are not justified by the data, and experiments could immensely benefit from having been performed with next-generation sequencing as opposed to qPCR (e.g. NET-seq or polyA mRNA-seq). This would solve some uncertainty in the claims and make the study overall much more convincing. Lastly, the manuscript is written quite confusingly and some sections (e.g. HP1gamma, H3K9me) could be omitted, which in my opinion would make the paper more clear and focused. As is, I suggest the study is more suited for Nature Communications. With additional NGS experiments, the study could fly at NSMB.

Major points:

1. Throughout the study, most bar graphs contain only 2 replicates and there are no statistical tests used to show significance. This is confusing in examples such as Fig. 4B, DS1 (is this decrease significant or not compared to ctrl sample) and Fig. S3, DS2, binding site (T and NT appear different but are claimed to not be different in the text).
2. There are several experiments that I don't think add much to the manuscript, but rather confuse and detract from the main message and could be omitted or moved entirely to supplement. For instance, Fig 2, Fig 6C-D, Fig S2A. I think the stress-induced termination defect and possible role of XRN2 is relevant and worth including, but DRB, HP1gamma, H3K9me, and PolII mutants do not lead to any solid conclusions that add to the work.
3. I find the authors' main argument that directionality of the gRNA is important for termination to be convincing. But I think a stellar experiment that the authors could include would be to target a region with overlapping sense and antisense transcripts, and show that with the gRNA in one direction, the sense transcript only is highly expressed and the antisense suppressed; and then with the gRNA in the opposite direction, the reverse expression pattern is observed. This would allow researchers to restrict transcription in only one direction for these types of gene pairs, like a "valve" as claimed in the paper.
4. The authors harp on differentiating between a decrease in mRNA levels, suppression of downstream readthrough, and alternative polyA site usage. For instance, when the gRNA is located in the gene body upstream of the normal polyA site, the authors report total

mRNA levels for the transcript decrease (Fig. 4B, in2, UTR). Meanwhile, when the gRNA is downstream of the normal polyA site (Fig. 4B, DS1, DS2, DS3), they claim prevention of downstream readthrough. And finally, when the gRNA is just downstream of an alternative polyA site, the authors claim alternative polyadenylation. I think the simplest explanation that explains all 3 of these observations is simple transcription elongation termination/stalling at the gRNA roadblock. The simple RT-qPCR method used throughout the paper is insufficient to differentiate between these possibilities. For example, in Fig. 4B, the in2 gRNA would stall transcription before the region targeted by the qPCR primer is transcribed, so of course mRNA levels assayed at that region would appear lower; that doesn't mean that overall transcript levels are lower/degraded. If the authors want to claim overall mRNA levels are decreased, they need to employ qPCR primers upstream of the in2 site, and better yet perform NET/RNA-seq and Northern blot. Conversely, gRNAs DS1, DS2, and DS3 target downstream of the qPCR primer, so of course transcription stalling will not affect levels assayed at that site. Finally, in Fig. 5, the authors cannot differentiate between transcriptional readthrough into the downstream primer region, and true alternative polyA site usage without performing a polyA mRNAseq similar to Fig 1A.

5. All in all, this manuscript would greatly benefit from Northern blots and RNA-seq-type experiments that would substantiate their claims about diminished overall RNA levels (RNA-seq), vs alternative splicing or polyadenylation (polyA mRNAseq), vs premature termination (NET-seq).

Minor points:

1. Please quantify the Western blot bands in Fig. S2A as the claimed changes appear rather slight by eye alone.
2. Please explain why CPT treatment includes TMEM188 e2 but excludes tbx3 e2a, or show diagram for each gene separately to better understand.
3. The discussion about kinetics for the difference in effect of dCas9 on alternative splicing vs alternative polyadenylation is weak. PolII pausing due to roadblock can directly trigger PolII termination due to known mechanisms (e.g. XRN2 pathway), whereas splicing occurs concurrently with but is not directly dependent upon PolII movement.

Reviewer #2:

Remarks to the Author:

Zukher and colleagues systematically target dCas9 to various positions in a gene body (near the TSS, in the middle of the gene, or near the PAS/TES) to determine effects on gene expression levels and transcription termination. They show dCas9 mediates transcriptional pausing, and this can induce transcription termination via a torpedo mechanism. These effects are only observed with guide RNAs basepairing to the non-template strand, which explains effects that were confusing in prior literature. The authors further show that the ultimate outcome strongly depends on target site context. For example, placing a dCas9 block between alternative PAS increases usage of the PAS located upstream of the block. The experiments are overall largely clear and a good use of ChIP, RNA level measurements, perturbations of key regulatory factors, and existing structural data are included. I have one major question and the remainder are small details. Overall, this study provides important parameters for optimal placement of dCas9

that should be very useful for the field.

- (1) Multiple guides are always used simultaneously, but is this necessary? Can a single guide RNA have the same effects on pausing/termination?
- (2) There are times when the writing should be tightened up and made more precise.

Minor points:

- (1) P. 6: "To further characterize the CRISPRi system, its effect on transcription beyond the 3'-ends of genes was investigated." This sentence made me think that the sgRNAs near the TSS were going to be used for this analysis, which is not what was done. Please clarify language to make it easier to read.
- (2) Fig 1B: A ratio is shown, but does the absolute level of transcript quantified with the upstream primers change? I assume not but I can not tell from the way the data are shown.
- (3) Figure S2A should be quantified.
- (4) P.16/Figure S3: It is written that there is no difference in dCas9 ChIP signal for T vs NT but it is unclear if this conclusion is true for the DS2 region.
- (5) P.18: "No mRNA suppression was detected with dCas9 targeted downstream of the PAS". This language is imprecise as DS1 did have an effect, as the authors discuss on p.20. This needs to be adjusted.

Author Rebuttal to Initial comments

Response to reviewers

We are grateful to our two reviewers for their thoughtful and valuable comments about our paper. We have endeavored to address all their points. In particular we have added new NGS data (**new Fig. 5, Ext. Data Fig. 5, Fig. 6c, Ext. Data Fig. 6b-d**). To make way for these additional data figures we have removed the old **Fig. 2b** (DRB treatment) as we felt these data were peripheral. The text has also been significantly shortened, again to make space for the new NGS data descriptions. The new NGS has not only confirmed our original RT-qPCR analyses but has also uncovered new aspects of CRISPRi.

- 1) We show activation of a nested intronic gene, *EID3* (within the very long intron 2 of *TXNDR1*) when a CRISPRi block is placed upstream. This effect is likely caused by blocking transcriptional interference from the gene's normal upstream promoter and highlights another potentially widespread side-effect of CRISPRi.
- 2) We confirm and extend our data on CRISPRi modulation of alternative polyadenylation.

Reviewer 1 (reviewer comments in italics)*Major points:*

1. *Throughout the study, most bar graphs contain only 2 replicates and there are no statistical tests used to show significance. This is confusing in examples such as **Fig. 4B**, DS1 (is this decrease significant or not compared to ctrl sample) and **Fig. S3**, DS2, binding site (T and NT appear different but are claimed to not be different in the text).*

In our revised ms we have added statistical tests to the figures mentioned above and to others, also adding more replicates where necessary. All the data points shown derive from fully independent biological replicates (separate transfections performed on separate days). However, we did not perform more than two replicates in cases where data was closely replicated.

2. *There are several experiments that I don't think add much to the manuscript, but rather confuse and detract from the main message and could be omitted or moved entirely to supplement. For instance, **Fig 2**, **Fig 6C-D**, **Fig S2A**. I think the stress-induced termination defect and possible role of XRN2 is relevant and worth including, but DRB, HP1gamma, H3K9me, and PolIII mutants do not lead to any solid conclusions that add to the work.*

We have removed the DRB analysis from original **Fig. 2** (second half). However, we have retained the data in original **Fig. 6** (now **Fig. 7**) as we consider the different responses of alternative polyadenylation (APA) and alternative splicing (AS) to CRISPRi roadblocks (affects APA not AS) versus elongation speed (affects AS not APA) to be important.

3. *I find the authors' main argument that directionality of the gRNA is important for termination to be convincing. But I think a stellar experiment that the authors could include would be to target a region with overlapping sense and antisense transcripts, and show that with the gRNA in one direction, the sense transcript only*

is highly expressed and the antisense suppressed; and then with the gRNA in the opposite direction, the reverse expression pattern is observed. This would allow researchers to restrict transcription in only one direction for these types of gene pairs, like a “valve” as claimed in the paper.

We agree this would be an interesting experiment but it is unfortunately beyond the scope of our present study.

4. *The authors harp on differentiating between a decrease in mRNA levels, suppression of downstream readthrough, and alternative polyA site usage. For instance, when the gRNA is located in the gene body upstream of the normal polyA site, the authors report total mRNA levels for the transcript decrease (Fig. 4B, in2, UTR). Meanwhile, when the gRNA is downstream of the normal polyA site (Fig. 4B, DS1, DS2, DS3), they claim prevention of downstream readthrough. And finally, when the gRNA is just downstream of an alternative polyA site, the authors claim alternative polyadenylation. I think the simplest explanation that explains all 3 of these observations is simple transcription elongation termination/stalling at the gRNA roadblock. The simple RT-qPCR method used throughout the paper is insufficient to differentiate between these possibilities. For example, in Fig. 4B, the in2 gRNA would stall transcription before the region targeted by the qPCR primer is transcribed, so of course mRNA levels assayed at that region would appear lower; that doesn't mean that overall transcript levels are lower/degraded. If the authors want to claim overall mRNA levels are decreased, they need to employ qPCR primers upstream of the in2 site, and better yet perform NET/RNA-seq and Northern blot. Conversely, gRNAs DS1, DS2, and DS3 target downstream of the qPCR primer, so of course transcription stalling will not affect levels assayed at that site. Finally, in Fig. 5, the authors cannot differentiate between transcriptional readthrough into the downstream primer region, and true alternative polyA site usage without performing a polyA mRNAseq similar to Fig 1A*

We hope that the RT-qPCR data are now presented in a clearer way. In particular Fig. 4B (Fig. 4c in the revised manuscript) and Fig. 5 (Fig. 6b in the revised

manuscript) bar charts show oligo-dT RT-qPCR with the PCR product over splice junctions, therefore measuring mature mRNA.

We also show Chr RNA and polyA⁺ RNA NGS data (**new Fig. 5 and Extended data Fig. 5**) that address these issues. In particular the difference between nascent and polyA⁺ RNA is evident. Also, differential expression analysis confirms mRNA suppression in CRISPRi cells, as we originally observed with oligo-dT RT-qPCR data. We also demonstrate a transcription interference effect, activating an intronic nested independent *EID3* transcription unit.

5. *All in all, this manuscript would greatly benefit from Northern blots and RNAseq-type experiments that would substantiate their claims about diminished overall RNA levels (RNA-seq), vs alternative splicing or polyadenylation (polyA mRNAseq), vs premature termination (NET-seq).*

We consider our new NGS dataset fully addresses these concerns.

Minor points:

1. *Please quantify the Western blot bands in Fig. S2A as the claimed changes appear rather slight by eye alone.*

We have done this (see **Extended Data Fig. 2a** of the revised manuscript)

2. *Please explain why CPT treatment includes *TMEM188 e2* but excludes *tbx3 e2a*, or show diagram for each gene separately to better understand.*

We have clarified this in the text.

3. *The discussion about kinetics for the difference in effect of dCas9 on alternative splicing vs alternative polyadenylation is weak. PolII pausing due to roadblock can directly trigger PolII termination due to known mechanisms (e.g.*

XRN2 pathway), whereas splicing occurs concurrently with but is not directly dependent upon PolII movement.

We edited the text to clarify relationships between alternative splicing and Pol II kinetics.

Reviewer 2

Major points:

(1) Multiple guides are always used simultaneously, but is this necessary? Can a single guide RNA have the same effects on pausing/termination?

We used several guides in most of our experiments to ensure a strong on-target effect. However, example of a single guide, changing *CCND1* polyA usage, suggests that a single guide is sufficient for a detectable effect. It would be interesting to address this and compare the effect of individual vs multiple targeting in future research.

(2) There are times when the writing should be tightened up and made more precise.

We have extensively edited the manuscript to hopefully address these concerns.

Minor points:

(1) P. 6: "To further characterize the CRISPRi system, its effect on transcription beyond the 3'-ends of genes was investigated." This sentence made me think that the sgRNAs near the TSS were going to be used for this analysis, which is not what was done. Please clarify language to make it easier to read.

The text has been edited in the revised manuscript to avoid this confusion.

(2) **Fig 1B**: *A ratio is shown, but does the absolute level of transcript quantified with the upstream primers change? I assume not but I cannot tell from the way the data are shown.*

For this experiment (**Fig. 1c** in the revised manuscript) we aimed to estimate transcription readthrough, which is independent of absolute transcript levels.

We performed absolute transcript level quantifications for CRISPRi-KRAB as well but did not detect significant changes in levels of unspliced *TXNRD1* or *AGFG1*. Surprisingly, levels of unspliced *THOC2* decreased upon CRISPRiKRAB targeting. We suggest this is due to specific effect of the H3K9me3 mark on this gene.

(3) *Figure S2A should be quantified.*

Done (see **Extended data Fig. 2a** in the revised manuscript)

(4) *P.16/Figure S3: It is written that there is no difference in dCas9 ChIP signal for T vs NT but it is unclear if this conclusion is true for the DS2 region.*

The figure is now updated to include the significance test (**Fig. 3c** in the revised manuscript).

(5) *P.18: “No mRNA suppression was detected with dCas9 targeted downstream of the PAS”. This language is imprecise as DS1 did have an effect, as the authors discuss on p.20. This needs to be adjusted.*

We edited the text to clarify this issue.

Decision Letter, first revision:

Message: Our ref: NSMB-A46487A

29th Mar 2023

Dear Dr. Proudfoot,

Thank you for submitting your revised manuscript "Elongation roadblocks mediated by dCas9 across human genes modulate transcription and nascent RNA processing" (NSMB-A46487A). It has now been seen by the original referees and their comments are below. The reviewers find that the paper has improved in revision, and therefore we'll be happy in principle to publish it in Nature Structural & Molecular Biology, pending minor revisions to satisfy the referees' final requests and to comply with our editorial and formatting guidelines.

To facilitate our work at this stage, it is important that we have a copy of the main text as a word file. If you could please send along a word version of this file as soon as possible, we would greatly appreciate it; please make sure to copy the NSMB account (cc'ed above).

Sincerely,

Carolina Perdigoto, PhD
Chief Editor
Nature Structural & Molecular Biology
orcid.org/0000-0002-5783-7106

Reviewer #1 (Remarks to the Author):

The authors have addressed all my scientific concerns and have tidied up the language of the manuscript to make it much more easily understood. I still think it would be cool to include the experiment I suggested targeting overlapping antisense transcripts and showing a one-way roadblock depending on the directionality of the gRNAs used. Overall, I recommend this work for publication in NSMB. Here are some final minor comments:

Minor comments:

- Abstract could use a sentence of introduction before jumping right into the findings to frame the importance of the work.
- "TSS" is mentioned in the Introduction without first defining it as "transcription start site".

- The intro is quite short and could benefit from some background discussion of Pol II termination, readthrough, processing, etc.
- Fig1c, indicate region targeted by gRNAs (currently just says "1-2 kb").
- ExtFig1b, typo: "downstream form PAS".
- Missing period at end of first results section.
- It would be useful to show one extended data figure comparing use of a single gRNA vs. your set of multiple gRNAs for a gene to see how many are needed for the effect.
- Pol II "NTD" is mentioned in the third results section but not defined until afterwards in the next section.
- All in all, Fig4 makes good sense. Targeting within the gene unit (intron, UTR) leads to pausing/premature termination before the PAS and thus mRNA decrease. Targeting outside the gene unit (after PAS) leaves the transcript intact but just alters the termination site, which is somewhat variable anyway and should not have much consequence. It is slightly surprising DS1 alters mRNA levels, but maybe you can comment that because it is so close to the PAS (although downstream) it may sterically interfere with proper cleavage and polyadenylation and thus decrease mRNA levels?
- The new Fig5 data are fantastic and the unexpected Eid3 finding is very cool and supports the proposed mechanism! I think it is really worth nailing the point though in the text by drawing attention to the subtle difference between the chr RNA and polyA RNA-seq data. For the chr RNA data, there is only a decrease AFTER the gRNA site (as shown nicely by the heat map below the tracks). And this correlates with co-transcriptional blockage. But for the polyA data, there is a decrease transcript-wide in the mature mRNA (for in2), indicating mature transcript instability due to abortive transcription before it can be properly terminated/polyadenylated (as shown nicely by the DE analysis). This is what I was looking for when I suggested it. Although it does appear in Fig5c that the 3' end of the DS2 sample is somewhat lower than the CTRL, though this wasn't picked up in the DE analysis. Can the authors comment briefly?
- New Fig6 data is also very nice. Maybe highlight in another color the alternative polyA sites like you do for the gRNAs.

Reviewer #2 (Remarks to the Author):

The authors have done a nice job addressing prior concerns through the addition of new data and clarifying the text/figures. The manuscript overall provides important insights into how dCas9 affects gene expression when guide RNAs are targeted at a variety of locations in and around a locus. It should serve as an important guide for researchers trying to use and interpret results obtained with dCas9.

The GEO submission number needs to be included in the final accepted version.

Minor text suggestion:

I feel the abstract should be adjusted to include the experimental approach used, e.g. at the beginning of the second sentence. This will help distinguish what information is background vs what was found in the current work.

Author Rebuttal, first revision:

Response to reviewers

Reviewer #1 (reviewer comments in italics)

The authors have addressed all my scientific concerns and have tidied up the language of the manuscript to make it much more easily understood. I still think it would be cool to include the experiment I suggested targeting overlapping antisense transcripts and showing a one-way roadblock depending on the directionality of the gRNAs used. Overall, I recommend this work for publication in NSMB. Here are some final minor comments:

Minor comments:

- Abstract could use a sentence of introduction before jumping right into the findings to frame the importance of the work.

Introduction sentence is now added.

- "TSS" is mentioned in the Introduction without first defining it as "transcription start site".

fixed

- The intro is quite short and could benefit from some background discussion of Pol II termination, readthrough, processing, etc.

Introduction is now expanded by adding a brief first paragraph about RNA polymerase II nascent and associated RNA processing

- Fig1c, indicate region targeted by gRNAs (currently just says "1-2 kb").

done

- *ExtFig1b, typo: “downstream form PAS”.*

done

- *Missing period at end of first results section.*

done

- *It would be useful to show one extended data figure comparing use of a single gRNA vs. your set of multiple gRNAs for a gene to see how many are needed for the effect.*

We didn't perform a full side to side comparison to generate a proper figure. Our preliminary data show that different sgRNAs can be efficient in the roadblocking, but there can be variations in individual sgRNAs efficiencies and/or stabilities in the cell. We do not know the determinants for this variation, so for now we suggest using a cocktail of guides as this allows for more robust effect with no need to test each individual sgRNA.

- *Pol II “NTD” is mentioned in the third results section but not defined until afterwards in the next section.*

edited the text re Fig.3b

- *All in all, Fig4 makes good sense. Targeting within the gene unit (intron, UTR) leads to pausing/premature termination before the PAS and thus mRNA decrease. Targeting outside the gene unit (after PAS) leaves the transcript intact but just alters the termination site, which is somewhat variable anyway and should not have much consequence. It is slightly surprising DS1 alters mRNA levels, but maybe you can comment that because it is so close to the PAS (although downstream) it may sterically interfere with proper cleavage and polyadenylation and thus decrease mRNA levels?*

We agree with the reviewer in this interpretation. We also think DS1 block has this unexpected effect on the mRNA as it induces Pol II pausing very close to the PAS and this can obstruct proper cleavage and polyadenylation, even though the pausing occurs downstream of the gene body.

- *The new Fig5 data are fantastic and the unexpected Eid3 finding is very cool and*

supports the proposed mechanism! I think it is really worth nailing the point though in the text by drawing attention to the subtle difference between the chr RNA and polyA RNA-seq data. For the chr RNA data, there is only a decrease AFTER the gRNA site (as shown nicely by the heat map below the tracks). And this correlates with co-transcriptional blockage. But for the polyA data, there is a decrease transcript-wide in the mature mRNA (for in2), indicating mature transcript instability due to abortive transcription before it can be properly terminated/polyadenylated (as shown nicely by the DE analysis). This is what I was looking for when I suggested it. Although it does appear in Fig5c that the 3' end of the DS2 sample is somewhat lower than the CTRL, though this wasn't picked up in the DE analysis. Can the authors comment briefly?

There is indeed an apparent slight decrease of *TXNRD1* expression in DS2 cells on the screenshot from Fig. 5c. It can also be seen in the normalized counts for replicate 1 shown at the screenshot, but it is not reproduced in the other replicates (Extended Data Fig. 5, compare rep1 and reps2-3 CTRL vs DS2). Hence we consider this apparent mRNA suppression to be insignificant, coming from a technical variation rather than from a real biological effect.

- New Fig6 data is also very nice. Maybe highlight in another color the alternative polyA sites like you do for the gRNAs.

We changed PAS presentation on Fig6 and ExtFig6.

Reviewer #2 (reviewer comments in italics):

The authors have done a nice job addressing prior concerns through the addition of new data and clarifying the text/figures. The manuscript overall provides important insights into how dCas9 affects gene expression when guide RNAs are targeted at a variety of locations in and around a locus. It should serve as an important guide for researchers trying to use and interpret results obtained with dCas9.

The GEO submission number needs to be included in the final accepted version.
done

Minor text suggestion:

I feel the abstract should be adjusted to include the experimental approach used, e.g. at the beginning of the second sentence. This will help distinguish what information is background vs what was found in the current work.

Unfortunately, due to the tight word limit we are unable to describe experimental approach in the abstract. However, we have tried to distinguish background knowledge from new finding in the now extended introduction section.

Final Decision Letter:

Message Dear Dr Zukher,

:

Please find below a copy of the decision letter for your manuscript "Elongation roadblocks mediated by dCas9 across human genes modulate transcription and nascent RNA processing" [NSMB-A46487B], which has just been accepted for publication in Nature Structural & Molecular Biology.

The exact publication date will be communicated to the corresponding author. Please note that until publication, the content of your paper remains under embargo (to determine when the paper can be discussed with the media, please consult our embargo policy at http://www.nature.com/authors/editorial_policies/embargo.html).

As soon as your article is published, you can generate your shareable link by entering the DOI of your article here: http://authors.springernature.com/share. Corresponding authors will also receive an automated email with the shareable link

If you wish to order reprints of your article or have any questions about reprints please send an email to author-reprints@nature.com.

Please contact the corresponding author directly with any queries you may have related to the content and publication of your paper.

As we prepare the manuscript for publication, we would like to confirm that your address details are correct. Could you please click on the link below to verify your profile and correct it as needed? Your prompt attention to this will help us to avoid delays in publication of your manuscript.

Please verify your address details promptly and correct them as needed by clicking here and following the link to "Login to My Account/Modify My NRG Profile":

[redacted]

Sincerely,
Carolina Perdigoto, PhD
Chief Editor
Nature Structural & Molecular Biology
orcid.org/0000-0002-5783-7106

Subject: Decision on Nature Structural & Molecular Biology submission NSMB-A46487B

Dear Dr. Proudfoot,

We are now happy to accept your revised paper "Elongation roadblocks mediated by dCas9 across human genes modulate transcription and nascent RNA processing" for publication as a Article in Nature Structural & Molecular Biology.

To assist our authors in disseminating their research to the broader community, our SharedIt initiative provides all co-authors with the ability to generate a unique shareable link that will allow anyone (with or without a subscription) to read the published article.

Recipients of the link with a subscription will also be able to download and print the PDF.

As soon as your article is published, you can generate your shareable link by entering the DOI of your article here: http://authors.springernature.com/share. Corresponding authors will also receive an automated email with the shareable link

Your paper will be published online soon after we receive proof corrections and will appear in print in the next available issue. You can find out your date of online publication by contacting the production team shortly after sending your proof corrections. Content is published online weekly on Mondays and Thursdays, and the embargo is set at 16:00 London time (GMT)/11:00 am US Eastern time (EST) on the day of publication. Now is the time to inform your Public Relations or Press Office about your paper, as they might be interested in promoting its publication. This will allow them time to prepare an accurate and satisfactory press release. Include your manuscript tracking number (NSMB-A46487B) and our journal name, which they will need when they contact our press office.

About one week before your paper is published online, we shall be distributing a press release to news organizations worldwide, which may very well include details of your work. We are happy for your institution or funding agency to prepare its own press release, but it must mention the embargo date and Nature Structural & Molecular Biology. If you or your Press Office have any enquiries in the meantime, please contact press@nature.com.

An online order form for reprints of your paper is available at https://www.nature.com/reprints/author-reprints.html. Please let your coauthors and your institutions' public affairs office know that they are also welcome to order reprints by this method.

Please note that Nature Structural & Molecular Biology is a Transformative Journal (TJ). Authors may publish their research with us through the traditional subscription access route or make their paper immediately open access through payment of an article-

processing charge (APC). Authors will not be required to make a final decision about access to their article until it has been accepted. [Find out more about Transformative Journals](https://www.springernature.com/gp/open-research/transformative-journals)

Authors may need to take specific actions to achieve [compliance with funder and institutional open access mandates](https://www.springernature.com/gp/open-research/funding/policy-compliance-faqs). If your research is supported by a funder that requires immediate open access (e.g. according to [Plan S principles](https://www.springernature.com/gp/open-research/plan-s-compliance)) then you should select the gold OA route, and we will direct you to the compliant route where possible. For authors selecting the subscription publication route, the journal's standard licensing terms will need to be accepted, including [self-archiving policies](https://www.springernature.com/gp/open-research/policies/journal-policies). Those licensing terms will supersede any other terms that the author or any third party may assert apply to any version of the manuscript.

Sincerely,

Carolina Perdigoto, PhD
Chief Editor
Nature Structural & Molecular Biology
orcid.org/0000-0002-5783-7106

Click here if you would like to recommend Nature Structural & Molecular Biology to your librarian:

<http://www.nature.com/subscriptions/recommend.html#forms>